# Complementarity-determining region clustering may cause CAR-T cell dysfunction

Tina Sarén [1], Giulia Saronio[1], Paula Marti Torrell[1], Xu Zhu[1], Josefin Thelander[1], Yasmin Andersson[2], Camilla Hofström[2], Marika Nestor[1], Anna Dimberg [1], Helena Persson[2], Mohanraj Ramachandran [1], Di Yu [1,3] ✉ & Magnus Essand [1,3] ✉

Chimeric antigen receptor (CAR)-T cell therapy is rapidly advancing as cancer treatment, however, designing an optimal CAR remains challenging. A single-chain variable fragment (scFv) is generally used as CAR targeting moiety, wherein the complementarity-determining regions (CDRs) define its specificity. We report here that the CDR loops can cause CAR clustering, leading to antigen-independent tonic signalling and subsequent CAR-T cell dysfunction. We show via CARs incorporating scFvs with identical framework and varying CDR sequences that CARs may cluster on the T cell surface, which leads to antigen-independent CAR-T cell activation, characterized by increased cell size and interferon (IFN)-γ secretion. This results in CAR-T cell exhaustion, activation-induced cell death and reduced responsiveness to target-antigen-expressing tumour cells. CDR mutagenesis confirms that the CAR-clustering is mediated by CDR-loops. In summary, antigen-independent tonic signalling can be induced by CDR-mediated CAR clustering, which could not be predicted from the scFv sequences, but could be tested for by evaluating the activity of unstimulated CAR-T cells.

Therapies based on chimeric antigen receptor (CAR)-T cells have revolutionized the treatment of blood cancers[1–4] and it has become evident that the design of the CAR construct is critical for optimal function and persistence of CAR-T cells in cancer patients. The single-chain variable fragment (scFv), responsible for antigen-recognition, the linker, hinge, transmembrane, and cytoplasmic signalling domains of the CAR can all have a substantial impact on the activation, proliferation, target-cell killing, and exhaustion of CAR-T cells[5]. A number of CARs have are known elicit chronic activation in the absence of ligand, exogenous cytokines, or feeder cells[6]. This has been associated with specific characteristics of the individual scFv used in the CAR constructs, with CAR clustering on the surface of T cells leading to antigen-independent tonic signalling[6–8]. Of note, CD19-targeting scFv derived from clone FMC63, which is used in clinically approved CAR-T cell products, appears not to induce tonic signalling[6,7], which can be a

contributing factor why CD19-targeted CAR-T cells have been so successful against B cell malignancies. Prevention of excessive tonic signalling enhances CAR-T cell function and prevent CAR-T cells from undergoing rapid exhaustion[9]. This implies that CARs lacking or showing low levels of tonic signalling are advantageous for clinical development.

A scFv is a fusion protein containing the antibody variable regions of the heavy ($V_H$) chain and light ($V_L$) chain, connected via a peptide linker. The $V_H$ and $V_L$ each contain three antigen-interacting complementarity-determining regions (CDRs) with framework regions in between, both of which can affect the stability and biophysical properties of scFvs[10,11]. It is known that the framework sequence significantly impacts the structural stability of the scFv, which might cause CAR clustering and altered CAR surface expression[12] and subsequently induce tonic signalling[7]. Replacing the framework structures

[1]Uppsala University, Dept Immunology, Genetics, Pathology, Science for Life Laboratory, Uppsala, Sweden. [2]Royal Institute of Technology (KTH), Drug Discovery and Development Platform, Science for Life Laboratory, Solna, Sweden. [3]These authors contributed equally: Di Yu, Magnus Essand. ✉e-mail: di.yu@igp.uu.se; magnus.essand@igp.uu.se

in the scFv through CDR-grafting can improve the scFv structural stability and CAR expression efficacy[12]. So far, the CDR sequence of the CAR has not been implied in tonic signalling.

In this study we develop several CARs that share the same framework regions and only differ in their CDR loops. We find that differences in the amino acid sequences of the CDR loops can profoundly impact CAR clustering resulting in antigen-independent activation, so called tonic signalling, which results in impaired CAR-T cell function. This data suggests that empirical screening for tonic signalling is important when developing new CARs.

## Results

### IL13Rα2-directed CAR-T cells display varied activity against target cells

Five scFv molecules [A]-[E] were selected by phage display panning (Fig. 1a) based on specific binding to human IL13Rα2 (Fig. 1b, Fig. S1a−h) but not the more abundantly expressed IL13Rα1 (Fig. S1i). Overall, the thermal stability of the scFvs were similar, although slightly lower for scFv[B] (Fig. 1b). We found that scFv[A], [B], and [C] bound in the ligand-binding domain of IL13Rα2, as binding could be blocked by the addition of excess IL-13, which was not the case for scFv[D] and [E] (Fig. S1j−s). The scFvs shared the same framework regions and differed only by a few amino acids in the complementarity-determining regions (CDRs). Five IL13Rα2-specific CAR-T cell constructs were engineered using lentiviral vectors encoding a second-generation CAR molecule consisting of the scFv connected to intracellular signalling domains of 4-1BB and CD3ζ (Fig. 1c). A lentivirus without a CAR was used to generate Mock-Ts. All constructs also encoded green fluorescent protein (GFP) as a reporter

of lentiviral transduction efficiency (Fig. 1c). Once expressed on T cells all five CARs bound to soluble IL13Rα2 (Fig. S2A) at levels matching the affinity of the corresponding scFv (Fig. 1b). When evaluated in vitro (Fig. 1d), all five CAR-Ts efficiently killed the human glioblastoma cell line U-87MG (Fig. 1e), which express high levels of IL13Rα2 (Fig. S2b−c). CAR[D]-T, CAR[E]-T and to some extent CAR[C]-T also showed potent cytotoxicity against glioblastoma cell lines expressing lower levels of IL13Rα2 (Fig. 1f; Fig. S2b−e) and secreted IFN-γ in a dose-dependent manner (Fig. 1g, h; Fig. S2f−g) and IL-2 (Fig. S2h−i). This was not the case for CAR[A]-T and CAR[B]-T (Fig. 1f−h; Fig. S2b−i). Importantly, none of the CAR-Ts displayed any cytotoxicity (Fig. S2J) nor secreted IFN-γ (Fig. S2k) in response to IL13Rα2-negative Mel526 melanoma cells. In addition, CAR[C]-T, CAR[D]-T and CAR[E]-T efficiently proliferated, as observed by the number of cell divisions, upon stimulation with U-87MG tumour cells, whereas CAR[A]-T and CAR[B]-T had impaired proliferative capacity (Fig. 1i). Furthermore, CAR[D]-T and especially CAR[E]-T delayed intracranial glioma growth and prolonged survival of mice (Fig. S3a−f). In conclusion, CAR[C]-T, CAR[D]-T and especially CAR[E]-T showed superior functionality and potency compared to CAR[A]-T and CAR[B]-T.

### Unstimulated CAR[A]-T and CAR[B]-T, but not CAR[E]-T cells, display tonic signalling due to CAR clustering

To determine the cause for the differing functionality of the CAR-Ts we compared the poorly functional CAR[A]-T and CAR[B]-T with highly functional CAR[E]-T at resting stage, i.e., without any antigen stimulation (Fig. 2a). The poor function of CAR[A]-T and CAR[B]-T could partially be explained by lower CAR surface expression compared to CAR[E]-T (Fig. 2b). In addition, CAR[A] and CAR[B] surface expression

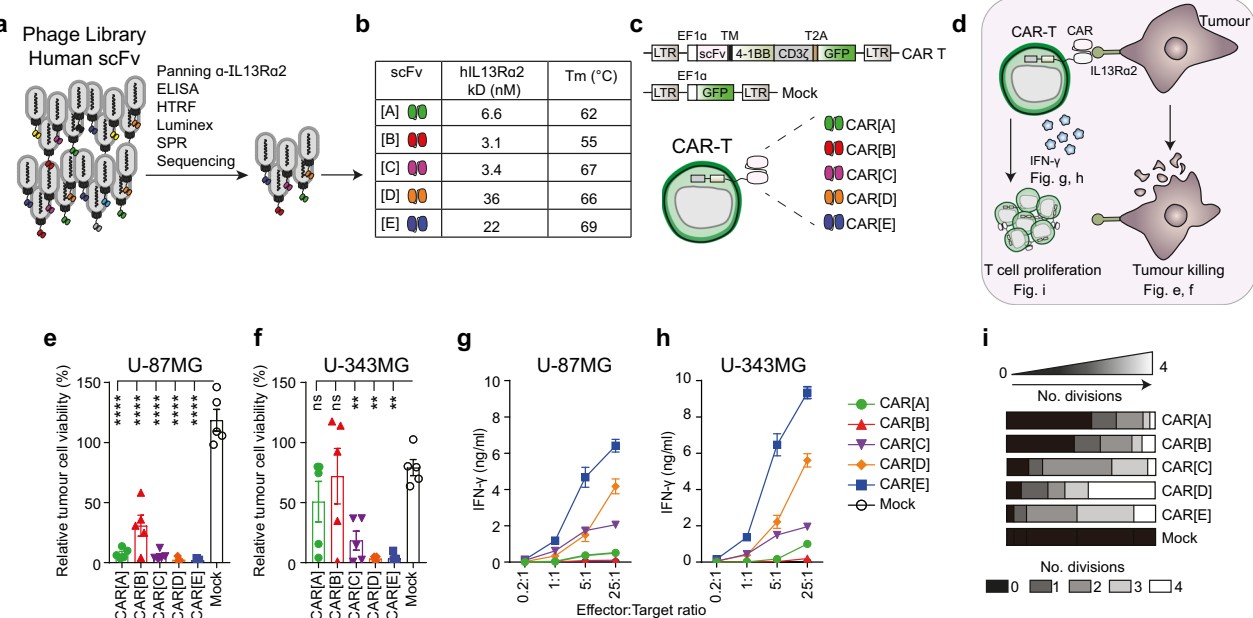

**Fig. 1 | IL13Rα2-directed CAR[C], CAR[D] and CAR[E]-T cells display better anti-tumour effect than CAR[A] and CAR[B]-T cells. a** Screening process to select single-chain variable fragments (scFv) targeting human IL13Rα2. **b** The affinity of selected scFv [A]-[E] against human IL13Rα2 as determined by Surface Plasmon Resonance (SPR). The melting temperature (Tm) in centigrade of scFv[A]-[E] was assessed to determine scFv stability. **c** Lentiviral constructs used to generate CAR[A]-[E]-Ts. Green fluorescent protein (GFP) was included for CAR-T detection and Mock-T, only encoding GFP, was included as control. **d** Illustration of experiments to evaluate CAR-T function. Experiments were performed after rapid expansion of the CAR-T cells on day 20 after T cell transduction. **e, f** Relative viability of (**e**) U-87MG and (**f**) U-343MG cells after 4d co-culture with CAR-Ts (25:1 Effector:Target cell ratio). Each dot represents T cells isolated from one healthy donor (n = 5) and data is

presented as bars of mean ± SEM. **e** p(CAR-Ts vs. Mock) < 0.0001, **f** p(CAR[C] vs. Mock) = 0.009, p(CAR[D] vs. Mock) = 0.0011, p(CAR[D] vs. Mock) = 0.0011. One-way ANOVA with Dunnett's correction for multiple comparison was used to compare between selected groups (**p ≤ 0.01, ****p ≤ 0.0001). **g, h** IFN-γ levels from 4d co-culture of CAR-Ts with (**g**) U-87MG or (**h**) U-343MG. T cells were isolated from healthy donors (n = 6) and data is shown as mean ± SEM. **i** Proliferation of CAR-Ts and Mock-T (gated as CD3+GFP+) after 4d co-culture with U-87MG, represented as the number of cell divisions. Data are shown as mean of experimental and biological duplicate. For all graphs in this figure: green circle: CAR[A]; red triangle: CAR[B]; purple reverse triangle: CAR[C]; orange diamond: CAR[D]; blue square: CAR[E]; empty circle: Mock. Source data are provided as Source Data file. EF1α: elongation factor 1 alpha, TM: transmembrane domain, T2A: self-cleaving peptide, No.: number.

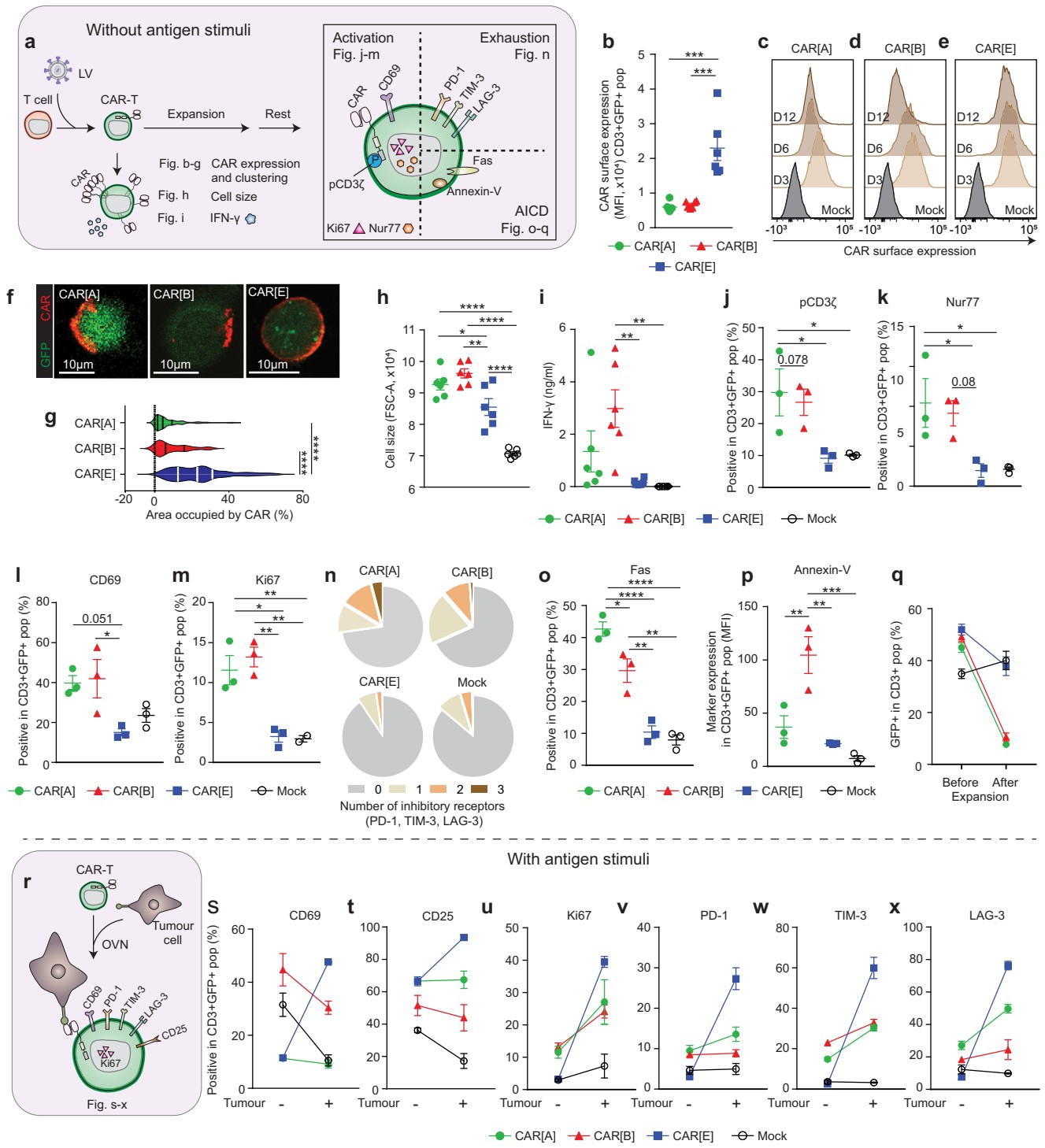

decreased over time in contrast to CAR[E] (Fig. 2c–e) suggesting instability of these CAR molecules. In line with this, out of the total CAR expression (intracellular and extracellular) significantly less of CAR[A] and especially CAR[B] molecules were located on the T cell surface compared to CAR[E] molecules (Fig. S4a). In addition, CAR[A] and CAR[B] molecules clustered on the T cell surface (Fig. 2f, g). Early on after transduction, CAR[A]-T and CAR[B]-T were larger in cell size (Fig. 2h) and secreted more IFN-γ than CAR[E]-T (Fig. 2i), indicating that CAR surface clustering induced a more activated phenotype. Activation in the absence of antigen-specific stimuli, suggested antigen-independent tonic signalling in CAR[A]-T and CAR[B]-T. Therefore, CAR-Ts were characterized for aspects associated with

tonic signalling and activation (Fig. 2a). CAR[A]-T and CAR[B]-T exhibited activating features including higher degree of phosphorylated CD3z (pCD3z) (Fig. 2j), upregulation of transcription factor Nur77 (Fig. 2k), higher surface expression of early activation marker CD69 (Fig. 2l), and higher proliferation (Ki67) (Fig. 2m) than CAR[E]-T without antigen-stimuli. Constitutive activation through CAR clustering induced phenotypical exhaustion of CAR[A]-T and CAR[B]-T in terms of expression of the inhibitory molecules PD-1, TIM-3, LAG-3 (Fig. 2n). In addition, antigen-independent activation of CAR[A]-T and CAR[B]-T resulted in activation-induced cell death as evidenced by higher expression of Fas in both CAR[A]-T and CAR[B]-T (Fig. 2o), Annexin-V for CAR[B]-T (Fig. 2p), and a decrease in percentage of both

**Fig. 2 | CAR[A]-T and CAR[B]-T, but not CAR[E]-T cells, become dysfunctional due to antigen-independent tonic signalling induced by CAR clustering.**
**a** Schematic illustration of the experimental set-up used to evaluate the CAR-T phenotype at a resting state, without antigen stimuli. **b** CAR surface expression on CAR-Ts (defined as CD3$^+$GFP$^+$) 3 days after T cell transduction. Each dot represents T cells isolated from one healthy donor ($n = 6$) and data is presented as mean ± SEM. p($_{CAR[A] vs. CAR[E]}$) = 0.0001, p($_{CAR[B] vs. CAR[E]}$) = 0.0002. One-way ANOVA with Tukey's correction for multiple comparison was used to compare between groups (***$p \leq 0.001$). **c–e** Representative histogram of the CAR expression level (shades of brown) on CD3$^+$GFP$^+$. **c** CAR[A]-T, **d** CAR[B]-T and **e** CAR[E]-T assessed 3, 6 and 12 days after transduction. Mock-T (grey) was used as control. **f** Representative confocal images showing the distribution of CAR molecules on the T cell surface and **g** quantified as the percentage of the T cell area occupied by CAR molecules (T cells generated from 3 healthy donors). Lines represent median (bold dotted) and quartiles (dotted). p($_{CAR[A] vs. CAR[E]}$) < 0.0001, p($_{CAR[B] vs. CAR[E]}$) < 0.0001. One-way ANOVA with Tukey's correction for multiple comparison was used to compare between all groups (****$p \leq 0.0001$). Experiments were performed on days 5–7 after T cell transduction. **h** The cell size (FSC-A) of CAR-Ts (gated as CD3$^+$GFP$^+$). p($_{CAR[A] vs. CAR[E]}$) = 0.04, p($_{CAR[A] vs. Mock}$) < 0.0001, p($_{CAR[B] vs. CAR[E]}$) = 0.002, p($_{CAR[B] vs. Mock}$) < 0.0001, p($_{CAR[E] vs. Mock}$) < 0.0001. **i** IFN-γ secreted from rested and unstimulated CAR-Ts. p($_{CAR[B] vs. CAR[E]}$) = 0.006, p($_{CAR[B] vs. Mock}$) = 0.004. Each dot represents T cells isolated from one healthy donor ($n = 6$) and data is presented as mean ± SEM. Experiments were performed on days 5–7 after T cell transduction. Flow cytometry analysis of (**j**) phosphorylated CD3z (pCD3z), (**k**) Nur77, (**l**) CD69 and (**m**) Ki67 expression in CAR-Ts (CD3$^+$GFP$^+$) without antigen stimuli. **j** p($_{CAR[A] vs. CAR[E]}$) = 0.038, p($_{CAR[A] vs. Mock}$) = 0.048, p($_{CAR[B] vs. CAR[E]}$) = 0.078. **k** p($_{CAR[A] vs. CAR[E]}$) = 0.039, p($_{CAR[A] vs. Mock}$) = 0.043, p($_{CAR[B] vs. CAR[E]}$) = 0.08. **l** p($_{CAR[A] vs. CAR[E]}$) = 0.051, p($_{CAR[B] vs. CAR[E]}$) = 0.035. **m** p($_{CAR[A] vs. CAR[E]}$) = 0.009, p($_{CAR[A] vs. Mock}$) = 0.01, p($_{CAR[B] vs. CAR[E]}$) = 0.0033, p($_{CAR[B] vs. Mock}$) = 0.005. Each dot represents T cells isolated from one healthy donor (**j–l**: $n = 3$; **m**: CAR-Ts: $n = 3$, Mock-T: $n = 2$) and data is presented as mean ± SEM. One-way ANOVA with Tukey's correction for multiple comparison was used to compare between all groups (*$p \leq 0.05$, **$p \leq 0.01$, ***$p \leq 0.001$, ****$p \leq 0.0001$). Experiments were performed after rapid expansion, on day 23 after T cell transduction. **n** Proportion of CAR-Ts (CD3$^+$GFP$^+$) expressing either 1, 2 or 3 inhibitory receptors (PD-1, TIM-3 and LAG-3) (CAR-Ts: $n = 3$, Mock-T: $n = 2$). The proportion of CD3$^+$GFP$^+$ cells expressing (**o**) Fas (%) and (**p**) Annexin-V (MFI). **o** p($_{CAR[A] vs. CAR[B]}$) = 0.025, p($_{CAR[A] vs. CAR[E]}$) < 0.0001, p($_{CAR[A] vs. Mock}$) < 0.0001, p($_{CAR[B] vs. CAR[E]}$) = 0.0027, p($_{CAR[B] vs. Mock}$) = 0.0012. **p** p($_{CAR[A] vs. CAR[B]}$) = 0.007, p($_{CAR[B] vs. CAR[E]}$) = 0.002, p($_{CAR[B] vs. Mock}$) = 0.0007. Each dot represents T cells isolated from one healthy donor ($n = 3$) and data is presented as mean ± SEM. One-way ANOVA with Tukey's correction for multiple comparison was used to compare between all groups (**$p \leq 0.01$, ***$p \leq 0.001$, ****$p \leq 0.0001$). Experiments were performed after rapid expansion of CAR-T cells, on day 23 after T cell transduction. **q** Reduction of CAR-Ts in culture, before and after expansion. Mock-T was used as control. Data was generated using T cells isolated from healthy donors (before $n = 6$; after $n = 4$) and mean is presented ± SEM. **r** Illustration of the experimental set-up to evaluate marker expression in antigen-stimulated CAR-Ts. The expression of (**s**) CD69, (**t**) CD25, (**u**) Ki67, (**v**) PD-1, (**w**) TIM-3 and (**x**) LAG-3 on CAR-Ts (CD3$^+$GFP$^+$). Each dot represents T cells isolated from one healthy donor (**s–x**: CAR-TsCAR-Ts: $n = 3$, Mock-T: $n = 2$;) and data is presented as mean ± SEM. Experiments were performed after rapid expansion, on day 29 after T cell transduction. All antibodies used can be found in Supplementary Table S1. Statistical comparisons that are not shown between groups are non-significant. For all graphs in this figure: green circle: CAR[A]; red triangle: CAR[B]; blue square: CAR[E]; empty circle: Mock. AICD activation-induced cell death, MFI mean fluorescence intensity, OVN overnight. Source data are provided as Source Data file.

CAR[A]-T and CAR[B]-T in the total T cell culture after expansion (Fig. 2q), compared to CAR[E]-T and Mock-T. Antigen-independent tonic signalling was mediated through the intracellular signalling domains of the CAR (Fig. S5). This was shown as CAR[B] lacking the intracellular domains (CAR[B]d) remained aggregated on the T cell surface (Fig. S5a, b) but the CAR[B]d-T cells displayed reduced cell size (Fig. S5c), IFN-γ secretion (Fig. S5d) and expression of inhibitory and activation markers (Fig. S5e–j).

When the CAR-Ts were exposed to IL13Rα2-expressing tumour cells, i.e., stimulated with the CAR-target antigen (Fig. 2r), both CAR[A]-T and CAR[B]-T were less responsive compared to CAR[E]-T. This was shown by the ability of CAR[E]-T to upregulate activation markers CD69 (Fig. 2s) and CD25 (Fig. 2t), proliferation marker Ki67 (Fig. 2u), as well as antigen-mediated upregulation of PD-1, TIM-3 and LAG-3 (Fig. 2v–x). CAR[A]-T and CAR[B]-T barely responded to antigen-specific stimuli (Fig. 2s–x). Analyses of all five CAR-Ts are presented in Fig. S6a–t. In conclusion, CAR[A]-T and CAR[B]-T display unstable CAR expression and a higher level of CAR clustering, which led to antigen-independent T cell activation, exhaustion, activation-induced cell death, and non-responsiveness to antigen stimuli. In contrast, CAR[E]-T had stable CAR expression and a low degree of CAR clustering in a rested state and responded well to antigen-specific stimuli.

### Gene expression analysis reveals antigen-independent activation of CAR[B]-T with subsequent exhaustion and cell death

To determine the molecular mechanisms that cause the difference in CAR-T functionality we used gene expression analysis on CAR-expressing (sorted on GFP expression) T cells that had not been exposed to any antigen-specific stimuli to compare highly functional CAR[E]-T with poorly functional CAR[B]-T in a rested state (Fig. 3a). Principal component analysis revealed that CAR[E]-T gene expression is similar to Mock-T, while CAR[B]-T had a distinct gene expression profile (Fig. 3b). When comparing CAR[B]-T against CAR[E]-T and Mock-T, there were 136 differentially expressed genes (Fig. 3c, Supplementary Table S2, Supplementary Data file 1). Genes associated with T cell activation and effector function (*e.g.*, *TNFRSF9*, *TNFRSF4*,

*TNFRSF18*, *GZMB*, and *GZMK*), T cell inhibition and exhaustion (*e.g.*, *HAVCR2*, *ENTPD1*, *CD38*, and *TOX*), and genes related to apoptosis (*e.g.*, *CASP3* and *TNFRSF10B*) were significantly upregulated in CAR[B]-T compared to CAR[E]-T and Mock-T (Fig. 3c). This was further confirmed by pathway score analysis, where rested CAR[B]-T showed distinct gene expression signatures in comparison to either CAR[E]-T or Mock-T, whose gene signature scores were similar. Unstimulated CAR[B]-T had higher activation, exhaustion, apoptosis, cytokine, JAK/STAT, glutamine metabolism, glycolysis, lipid metabolism, and mTOR signalling pathway scores (Fig. 3d–g) and higher activation of the Notch, RBPJ, and NFκB transcription pathways compared to CAR[E]-T and Mock-T (Fig. 3h). In conclusion, the gene expression analysis confirmed that CAR[B]-T was activated and exhausted already in the absence of antigen-specific stimuli further showing that CAR[B]-T display antigen-independent tonic signalling. In contrast, CAR[E]-T had a non-activated gene expression profile, similar to Mock-T prior to antigen stimulation.

### Tonic signalling is caused by CDR-mediated CAR clustering

The scFvs used to create the different CARs are based on the phage display screening of a library consisting of human heavy chain variable gene *IGHV3-23* and kappa light chain variable gene *IGKV1-39*. Thus, they have identical scFv frameworks and only differs by a few amino acids in CDR-1, −2 and −3 in the heavy (H) chain and CDR-3 in the light (L) chain. We performed alanine substitutions of the amino acids in the individual CDR loops of CAR[B] that differed from CAR[E] to determine if and in that case which of the CDR loops caused CAR[B] clustering, impaired CAR surface expression, and tonic signalling (Fig. 4a, b). Alanine substitutions either in CDR-2 of the heavy chain ([B]-H2) or CDR-3 of the light chain ([B]-L3) partially improved CAR surface expression in transduced T cells, while alanine substitution in all four CDR loops simultaneously ([B]-All) further improved CAR surface expression of CAR[B] on transduced T cells (Fig. 4c). Given the apparent importance of CDR-H2 and CDR-L3 for structural stability of CAR[B] we created a double mutant, with wild-type H2 and the most common L3 sequence ([B]-H2L3wt) of the germline sequences[13]

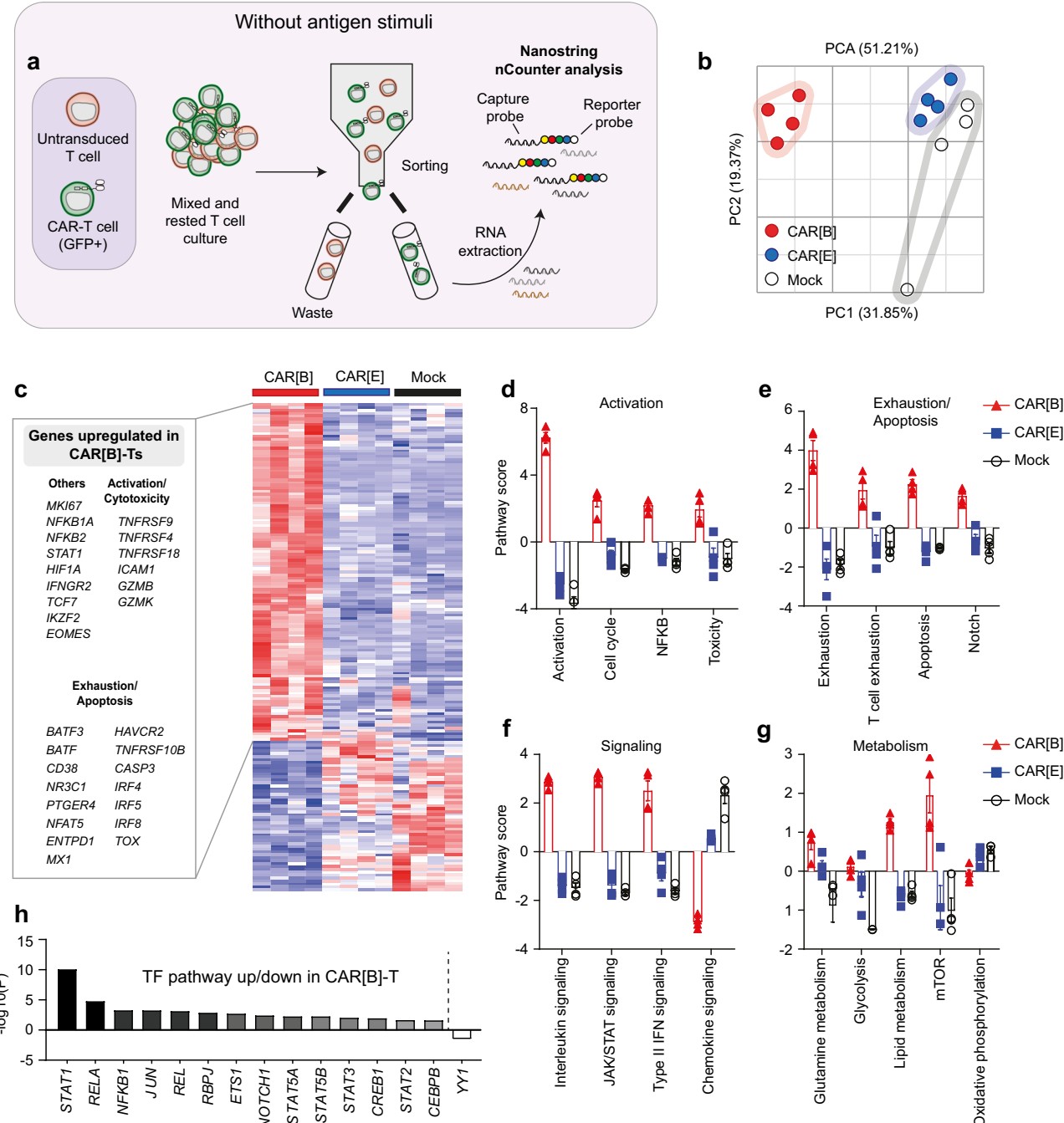

**Fig. 3 | Unstimulated CAR[B]-T, but not CAR[E]-T cells, exhibit an activated and exhausted gene expression profile. a** Summary of the sample preparation and subsequent gene expression, NanoString nCounter, analysis using the CAR-T characterization panel. The experiment was performed after rapid expansion followed by rest, on day 22 after T cell transduction. **b** Principal component analysis (PCA) of the transcriptional profile of CAR[B]-T, CAR[E]-T and Mock-T. Each dot represents T cells isolated from one healthy donor (n = 4). **c** Heatmap displaying the significantly (p ≤ 0.05) differentially expressed genes (DEGs) when comparing

CAR[B]-T vs. CAR[E]-T + Mock-T. DEG were determined using DESeq2 (Fold change < −1.49 or >1.49), (Supplementary Data file 1). Pathway scores of (**d**) activation, (**e**) exhaustion/apoptosis, (**f**) signalling and, (**g**) metabolism assigned using nSolver4.0 analysis. Data is presented as mean ± SEM (n = 4 in each group). **h** Transcription factor (TF) pathways significantly differentially regulated in CAR[B]-T based on DEGs (Supplementary Table S2). For all graphs in this figure: red triangle: CAR[B]; blue square: CAR[E]; empty circle: Mock. TF transcription factor. Source data are provided as Source Data file.

(Fig. 4b, lower panel). These substitutions improved CAR[B] surface expression to levels comparable with CAR[E] (Fig. 4d, Fig. S7a). Thus, our data indicated that the amino acid sequence in the CDR-H2 and CDR-L3 loops heavily affected the stability of the CAR[B] molecule. In line with this, the mutants [B]-H2, [B]-L3 and especially [B]-H2L3wt displayed reduced clustering on the T cell surface compared to CAR[B], *i.e.*, having a larger surface area occupied by CAR (Fig. 4e, f). As

a result, antigen-independent activation was significantly reduced, as shown by decreased cell size (Fig. 4g), IFN-γ secretion (Fig. 4h) and CD69 expression (Fig. 4i). Furthermore, the mutants [B]-H2, [B]-L3 and especially [B]-H2L3wt displayed reduced antigen-independent CAR-T exhaustion as evident by reduced number of inhibitory receptors compared to CAR[B]-T (Fig. 4j). As expected, some but not all CDR amino acid substitutions led to impaired binding strength to IL13Rα2.

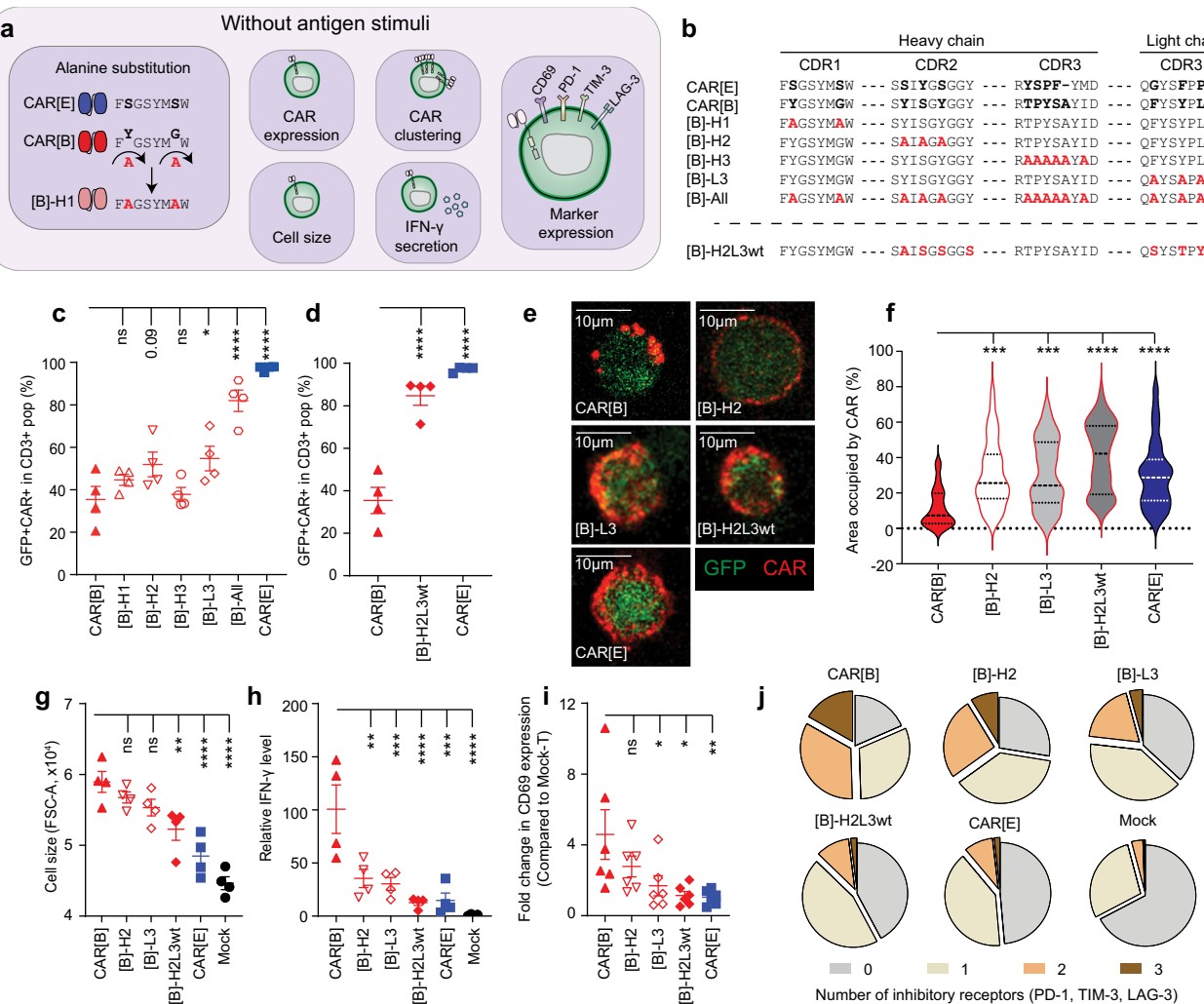

**Fig. 4 | Complementarity-determining region (CDR)-mediated CAR-clustering in non-stimulated CAR[B]-T cells induce tonic CAR-T signalling. a** Illustration of alanine substitutions and subsequent assays. **b** The variable amino acids (aa) (black bold) in the CDR regions between CAR[E] and CAR[B], and the corresponding constructs after alanine substitution (red bold). [B]-H2L3wt incorporated the wild-type H2 and the most commonly found L3 sequence of the original phage library (heavy chain variable gene IGHV3-23 and kappa light chain variable gene IGKV1-39). (H = Heavy chain, L = Light chain). **c, d** The proportion of CAR$^+$ cells in the transduced cell population (GFP$^+$), after engineering with **c** alanine mutants, or **d** wild type double-mutant. **c, d** p$_{(CAR[B] vs. [B]-H2)}$ = 0.086, p$_{(CAR[B] vs. [B]-L3)}$ = 0.035, p$_{(CAR[B] vs. [B]-All)}$ < 0.0001, p$_{(CAR[B] vs. CAR[E])}$ < 0.0001, p$_{(CAR[B] vs. [B]-H2L3wt)}$ < 0.0001. Each dot represents T cells isolated from one healthy donor ($n = 4$) and data is presented as mean ± SEM. **e** Representative images showing CAR molecule distribution on the T cell surface which was (**f**) quantified as T cell area occupied by CAR (T cells generated from 3 healthy donors). Lines represent median (bold dotted) and quartiles (dotted). p$_{(CAR[B] vs. [B]-H2)}$ = 0.0002, p$_{(CAR[B] vs. [B]-L3)}$ = 0.0001, p$_{(CAR[B] vs. [B]-H2L3wt)}$ < 0.0001, p$_{(CAR[B] vs. CAR[E])}$ < 0.0001. **g** Cell size (FSC-A) of transduced (CD3$^+$GFP$^+$) but unstimulated T cells. p$_{(CAR[B] vs. [B]-H2L3wt)}$ = 0.0064, p$_{(CAR[B] vs. CAR[E])}$ < 0.0001, p$_{(CAR[B] vs. Mock)}$ < 0.0001. **h** IFN-γ level secreted from

unstimulated CAR-T cells in the culture (normalized to fraction of transduced GFP$^+$ cells). p$_{(CAR[B] vs. [B]-H2)}$ = 0.0019, p$_{(CAR[B] vs. [B]-L3)}$ = 0.0009, p$_{(CAR[B] vs. [B]-H2L3wt)}$ < 0.0001, p$_{(CAR[B] vs. CAR[E])}$ < 0.0001, p$_{(CAR[B] vs. Mock)}$ < 0.0001. Each dot represents T cells isolated from one healthy donor ($n = 4$) and data is presented as mean ± SEM. **i** Fold change in the proportion of CAR-Ts (CD3$^+$GFP$^+$) expressing CD69 in comparison to Mock-T (CD3$^+$GFP$^+$). p$_{(CAR[B] vs. [B]-L3)}$ = 0.035, p$_{(CAR[B] vs. [B]-H2L3wt)}$ = 0.01, p$_{(CAR[B] vs. CAR[E])}$ = 0.008. Each dot represents T cells isolated from one healthy donor ($n = 6$) and data is presented as mean ± SEM. **j** Pie chart displaying the proportion of CD3$^+$GFP$^+$ cells expressing 0, 1, 2 or 3 inhibitory receptors (PD-1, TIM-3 and LAG-3). Data is shown as mean from T cells generated from 3 healthy donors. Antibodies used for staining can be found in Supplementary Table S1. One-way ANOVA with Dunnett's correction for multiple comparison was used to compare between selected groups (*$p ≤ 0.05$, **$p ≤ 0.01$, ***$p ≤ 0.001$, ****$p ≤ 0.0001$). All experiments in the figure were performed 5–7 days after T cell transduction. For all graphs in this figure: red triangle: CAR[B]; empty triangle: [B]-H1, empty reverse triangle: [B]-H2; empty circle: [B]-H3; empty diamond: [B]-L3, empty hexagon: [B]-All; red diamond: [B]-H2L3wt; blue square: CAR[E]; black circle: Mock. Source data are provided as Source Data file.

In fact, [B]-L3 had reduced tonic signalling, but maintained target binding (Fig. S7b) and responded slightly better to antigen stimulation (Fig. S7c) compared to CAR[B] (Fig. S7d–j). On the other hand [B]-H2L3wt, which did not display any tonic signalling, had completely lost binding to IL13Rα2 (Fig. S7b) and as a result did not respond to antigen stimulation (Fig. S7d–j). Similar results were also observed when mutating the CDRs of CAR[A] and analysing CAR[A]-T without exposure to IL13Rα2 (Fig. S8a, b). Alanine substitution revealed that in CAR[A] only mutation of CDR-H3 ([A]-H3) improved CAR surface

expression (Fig. S8c) and reduced clustering (Fig. S8d, e). As a result [A]-H3 displayed reduced tonic signalling as evident by reduced cell size (Fig. S8f) and autonomous IFN-γ secretion (Fig. S8g), but lost the binding capacity to IL13Rα2 (Fig. S8h). In conclusion, these data showed that the amino acid sequence in the CDR loops can heavily affect the stability of the CAR molecule, thus impact CAR clustering and subsequent antigen-independent activation and exhaustion.

To validate that CDR-mediated CAR clustering and subsequent antigen-independent tonic signalling is not unique to IL13Rα2-

targeting CARs we evaluated this phenomenon in CAR-Ts targeting the CD44v6 isoform (CAR[1] and CAR[2]) (Fig. S9a, b). The CD44v6-binding scFvs incorporated in the CARs were derived from the same phage display library as the IL13Rα2-biding scFv and thus share an identical framework and only differ in the CDRs. Although CAR[1] and CAR[2] only differed by a few amino acids in CDR-3 of the heavy and CDR-3 of the light chain, CAR[1] displayed impaired CAR surface expression (Fig. S9c), CAR clustering (Fig. S9d, e) and antigen-independent tonic signalling (Fig. S9f−m) in sharp contrast to CAR[2]. Alanine substitution of CAR[1] in positions differing from CAR[2] ([1]-H3 and [1]-L3) improved CAR surface expression (Fig. S10a), reduced CAR clustering (Fig. S10b, c) and antigen-independent tonic signalling (Fig. S10d−j), clearly demonstrating that also in this case CDR sequences were responsible for CAR clustering.

In conclusion, we show that the amino acids in the CDR loops can induce CAR clustering and subsequent antigen-independent tonic signalling in multiple CARs against different target antigens.

## Discussion

All elements of a CAR construct, including the extracellular scFv, linker, hinge, transmembrane and intracellular co-stimulatory domains can substantially impact the activation, proliferation, efficacy and exhaustion of CAR-T cells[14]. The CDR loops of the scFv are mainly responsible for antigen-recognition while the framework is often considered to impact the stability of the CAR and thus affect its expression and distribution on the cell membrane[12]. The framework of certain scFvs may cause a spontaneous antigen-independent signalling of CARs, leading to T cell dysfunction[7]. Herein we show that interaction between CDR loops of the scFv can also affect CAR stability and CAR-T functionality.

We isolated five scFvs, with varying affinity for the glioblastoma-associated antigen IL13Rα2, that recognized different epitopes of the receptor and developed CAR-T cells from the scFvs. All CAR-Ts killed glioblastoma cells with high levels of IL13Rα2 expression. However, killing of glioblastoma cells expressing lower levels of IL13Rα2 varied among the CAR-Ts. The poor response of CAR[A]-T and CAR[B]-T was associated with CAR aggregation and reduced expression of CAR molecules on the surface of T cells and antigen-independent tonic signalling. Antibody fragments and scFvs are prone to oligomerization[12,15,16] and in the context of CAR-Ts such oligomerization of scFvs leads to constitutive CAR signalling and T cell exhaustion[6,7]. In line with this, resting CAR[B]-T cells expressed elevated levels of genes involved in the development of an exhausted/dysfunctional cellular state. This development of CAR[B]-T dysfunction at resting state is similar to exhaustion/dysfunction of T cells in chronic viral infections and cancer[17−22]. This likely explains why CAR[A]-T and CAR[B]-T displayed an impaired response to antigen stimuli compared to CAR[E]-T. Removal of the intracellular signalling domain of CAR[B]-T prevented tonic signalling in the resting state despite oligomerization of CAR[B] on the T cell surface. Thus, it is clear that CAR clustering and subsequent tonic signalling is mediated through the intracellular signalling domain of CAR[B].

Although, CAR[A], CAR[B] and CAR[E] shared the same framework regions, and differs only by a few amino acids in the CDR loops, we observed that CAR[A] and CAR[B] are less stable, as they oligomerized, leading to reduced surface expression of CAR[A] and CAR[B] on the T cells over time. In addition, scFv[B] had a lower melting temperature (55 °C) compared to the other selected scFvs while scFv[A] had similar melting temperature as the other scFvs. This suggests that the CDR sequence and the scFv melting temperature cannot be used to predict CAR stability, clustering and subsequent tonic signalling. In line with this, the FMC63 derived scFv used in anti-CD19 CAR-T, which is not prone to cluster and tonic signalling[7] has a similar melting temperature (57.7 °C) as scFv[B][23]. Therefore, tonic signalling must thus be evaluated at the CAR-T level and cannot be predicted

from a library of recombinant scFvs. By mutating amino acids in the CDR loops of both IL13Rα2-targeting (CAR[A] and CAR[B]) and CD44v6-targeting (CAR[1]) CARs, CAR clustering and antigen-independent tonic signalling could be prevented, as evidenced by for example decreased cell size and lowered background IFN-γ secretion of unstimulated cells. These results clearly demonstrate that CDR loop sequences of multiple CARs, independent of CAR target antigen can induce antigen-independent tonic signalling which in turn can impair CAR-T functionality. It has previously been reported that a few amino acid differences in the scFv framework of a CAR may have significant impact on CAR stability and aggregation[12]. Our study shows that a few amino acid differences in the scFv CDR loops can also cause aggregation of CAR molecules. As small differences in various scFvs regions can cause CAR-clustering it can be challenging to predict oligomerization. Therefore, we propose a rapid screening method for CAR-clustering and tonic signalling in CAR-Ts using cell size and autonomous IFN-γ secretion. As these measures always had a positive correlation with CAR clustering and antigen-independent activation they could be used to select appropriate CARs for further development. In summary, we found that the CDR loops of the scFv in multiple CARs can significantly influence CAR stability, leading to clustering, induction of antigen-independent tonic signalling that ultimately lead to impaired CAR-T function.

## Methods

The research presented in this study complies with all relevant ethical regulations. The Northern Stockholm and Uppsala Research Animal Ethics Committee have approved all animal studies performed (N164/15, 5.8.18-19434-2019 and 5.8.18-13414.2020).

### Expression of human IL13Rα2

The extracellular domain of the human IL13Rα2 (Uniprot Q14627 aa 29-501) was co-expressed with BirA biotin ligase in the MultiBac Expression System (Geneva Biotech) to produce the biotinylated receptor. 5' of the receptor sequence, the gp67 baculovirus signal for secretion was incorporated while an Avi-tag followed by a His6-tag for affinity purification was added to the 3'end of the sequences. The construct was codon-optimized for *Spodoptera frugiperda*. Likewise, the *E. coli* BirA biotin ligase gene was codon-optimized. The genes were ordered from GeneArt Thermo Fisher Scientific. The multibac constructs of hIL13Rα2-AVIhis was cloned into the acceptor vector pACEBac1 and BirA was cloned into the donor vector pIDS. The vector sequences were verified by sequencing. The two vectors were fused forming the hIL13Rα2-AVIhis/BirA construct and the resulting construct was transformed into DH10EMBacY. Selection of positive bacmid clones were made by blue/white screening. Finally, the bacmid was isolated and analysed by PCR for incorporation of genes. The bacmid was transfected into Sf9 cells (ATCC, CRL-1711) to produce the baculovirus. The human IL13Rα2 was expressed in Sf9 transfected with the baculovirus during 48 h and harvested from the medium by capturing on HisTrap Excel column (Cytiva, 17371205). The column was equilibrated with buffer A (50 mM HEPES pH 7.0, 150 mM NaCl, 9 mM imidazole, 10% glycerol and 10 μM Tween). After washing the column with buffer A, the protein was eluted with Buffer B (50 mM HEPES pH 7.0, 150 mM NaCl, 9 mM imidazole, 10% glycerol, 10 μM Tween and 300 mM imidazole). After pooling and concentrating the protein containing fractions, the sample was polished on the Superdex 200 16/60 column (Cytiva, 28989335) using 50 mM HEPES pH 7.0, 150 mM NaCl, 10% glycerol and 10 μM Tween.

The purified hIL13Rα2 receptor was run on an SDS gel to determine purity and size. Binding to mouse anti-IL13Ra2 single chain clone 47 control[24] was verified by ELISA and Western blot. The receptor was sent to Xiaofang Cao, Clinical Proteomics Mass Spectrometry, Science for Life Laboratory (SciLifeLab, Stockholm, Sweden) for MS analysis to verify Protein ID.

## Phage display selection and scFv characterization

Phage display selections using the SciLifeLab synthetic library of human scFvs based on heavy chain variable gene *IGHV3-23* and kappa light chain variable gene *IGKV1-39* was constructed and designed as previously reported[13,25]. For the biotinylated human target (hIL13Rα2-avitag, described above), the selection was performed using streptavidin-coated magnetic beads (Dynabeads M-280 streptavidin, Invitrogen). Analogously, protein G-coupled magnetic beads were used to capture the Fc-fused mouse orthologue: mIL13Rα2-Fc (RnD Systems, #539-IR). In two of the selection tracks, in order to preferentially select for cross-species reactive scFv, the antigen was alternated between human and mouse IL13Rα2 in the different rounds. Furthermore, in another two tracks, human IL-13 ligand (Prospec, #cyt-446) was included in the selection buffer. It was reasoned that this might enrich for binders that bind to an epitope distinct from the IL-13 binding site, which may be advantageous. The selection pressure was increased by gradually decreasing the antigen amount (250, 100, 20 and 5 or 10 pmol, respectively) and by increasing the number and intensity of washes between the different rounds (5–8 washes). Elution of antigen-bound phages was performed using a trypsin-aprotinin approach. The entire selection process, except the phage-target protein incubation step, was automated and performed with a Kingfisher Flex robot (Thermo Fisher Scientific). Phagemid DNA from round 3 and 4 were purified and the scFv genes ($V_H$-linker-$V_L$) were transferred to an expression vector as described previously[25].

Following selections, cloning and transformation, a total of 920 colonies were picked from round 3 and 4 and analysed further. ELISA revealed 673 colonies producing true binders, of which 304 were identified as unique clones after DNA sequencing. The uniquely defined binders were included in further validation, which was performed as previously described including ELISA, Homogeneous Time Resolved Fluorescence (HTRF), Luminex, and Surface Plasmon Resonance (SPR)[25]. Based on these results eleven scFv candidates, were selected to be produced in larger scale and protein A purified as described[25] and tested in additional assays.

## More detailed characterization of a subset of selected scFv

Binding kinetics of the eleven selected scFv were determined by SPR using Biacore T200 instrument (GE Healthcare). An α-FLAG M2 antibody (Sigma-Aldrich #F1804, Supplementary Table S1), functioning as a capture ligand was immobilized onto all four surfaces of a CM5-S amine sensor chip on a Biacore T200 instrument (GE Healthcare) according to manufacturer´s recommendations. Protein-A-purified FLAG-tagged scFv clones were each injected and captured onto the chip surface. Human IL13Rα2-avitag protein (made in house) was sequentially injected over the chip surface (at a concentration ranging from 1.2–100 nM). Following a dissociation phase, the chip surfaces were regenerated with 10 mM glycine-HCl, pH 2.1. All experiments were performed at 25 °C in HBS supplemented with 0.05% Tween20, pH 7.5. By subtracting the response curve of a reference surface, being an α-FLAG M2 antibody immobilized surface, response curve sensorgrams for all scFv clones were obtained. Data was analysed using software BIAeval v.3.1 (GE Healthcare) and kinetic parameters were calculated assuming a 1:1 Langmuir binding model.

Cross-reactivity to IL13Rα1 was investigated by ELISA. Coating proteins hIL13Rα2-avi tag, hIL13Rα2-Fc (RnD systems, #7147-IR), mIL13Rα2-Fc (RnD Systems, #539-IR) and hIL13Rα1-Fc (RnD Systems, #146-IR) were diluted to 1 µg/ml in PBS and directly coated into a 384-well ELISA plate. Two negative control proteins, streptavidin and a non-relevant protein, were also coated. Following incubation of plates with coating-proteins overnight at 4 °C, plates were washed twice with MilliQ water and blocked for 2 h in blocking buffer (PBS supplemented with 0.5% BSA and 0.05% Tween20). FLAG-tagged scFv clones present in bacterial supernatant were diluted 1:10 in blocking buffer and allowed to bind. Detection of binding was enabled through an HRP-conjugated α-FLAG M2 antibody (Sigma-Aldrich, #A8592, Supplementary Table S1) followed by incubation with 1-step Ultra TMB-ELISA substrate (Thermo Fisher Scientific, #34029). The colorimetric-signal development was stopped by addition of 1 M sulfuric acid and plates were analysed at 450 mm.

An IL-13 competition assay was performed using SPR. The α-FLAG M2 antibody, was immobilized onto all four surfaces of a CM5-S amine sensor chip as described above. FLAG-tagged scFv clones were injected and captured onto the chip surfaces, followed by injection of either 100 nM hIL13Rα2-avitag or 100 nM hIL13Ra2-avitag pre-incubated with 200 nM IL-13 (Prospec #cyt-446). The surfaces were regenerated with 10 mM glycin-HCl pH 2.1.

The melting temperature was assessed as follows: The protein A purified scFv clones were diluted to 0.1 mg/ml in PBS and loaded into "high sensitivity" capillary (NanoTemper #PR-C006) through capillary force. Melting temperature ramp was set between 20 °C and 95 °C, heating 1 °C/min. Tryptophan emission was measured at 330 nm and 350 nm and the calculated ratio plotted against temperature to obtain melting curves for each clone, from which Tm-values can be deduced using software PR. ThermoControl (NanoTemper Inc).

## Lentivirus construction and production

Based on the characterization studies above, five scFv were selected as prime candidates, scFv [A], [B], [C], [D] and [E]. These were cloned into a cassette encoding CAR constructs containing the signalling domain of CD3zeta and the co-stimulatory signalling domain of CD137 as previously described[26]. The CAR cassette was cloned into a third-generation self-inactivating (SIN) lentiviral vector (SBI, System Biosciences) under the control of elongation factor-1 alpha (EF1a) promoter. Green fluorescent protein was incorporated after the CAR cassette and a self-cleaving T2A sequence. A mock construct only encoding GFP under the control of the EF1a promoter was used as a control. All sequences were purchased (Genscript). Production of third generation viral particles has been previously described[27]. Decoy variants were constructed by removing the intracellular domain, and the alanine mutations with corresponding sequences were ordered (Genscript) and constructed as described above.

## CD44v6-targeted CAR-Ts

ScFv against CD44v6 were developed by phage display selections using the SciLifeLab synthetic library of human scFvs based on heavy chain variable gene *IGHV3-23* and kappa light chain variable gene *IGKV1-39*[13,25]. The scFv sequences were cloned into the same lentiviral vector backbone and lentivirus particles were produced as described above for IL13Rα2-directed CARs.

## Culture conditions of cell lines and primary cells

The human glioblastoma cell lines U-87MG (UU), U-87MG (ATCC-HTB14), U-343MG (UU), U-251MG (UU) and were cultured in MEM (Minimum Essential Medium Eagle) supplemented with 10% Fetal Bovine serum (FBS), 1% penicillin/streptomycin (PEST) and 1% sodium pyruvate. In this paper both the original U-87MG Uppsala and U-87MG (ATCC) were used as they are both are likely glioblastoma cell lines and express IL13Rα2.

The human melanoma cell line Mel526 (Cellosaurus, VCVL_8051) was cultured in DMEM (Dulbecco's Modified Eagle Medium) Glutamax supplemented with 10% FBS, 1% PEST and 1% sodium pyruvate. THP-1 cells (TIB-202, ATCC) and Jurkat cells (D1.1, ATCC CRL-10915) were cultured in RPMI supplemented with 10% FBS, 1% PEST and 1% sodium pyruvate.

The cell lines used in this study were not authenticated. Human T cells were maintained in (RPMI 1640 medium supplemented with 10%FBS, 1% PEST, 1% sodium pyruvate and 50 IU/ml IL-2 (Proleukin, Novartis). All reagents were purchased from Thermo Fisher Scientific, unless mention differently.

## Jurkat cell engineering

Jurkat cells were transduced by mixing $2.5 \times 10^5$ cells (in 250 μl RPMI culture medium), with 10 μl concentrated lentivirus and 5 μg/ml polybrene. Cells were centrifuged at $800 \times g$ at 32 °C for 30 min. Cells were re-suspended in RPMI culture medium. Two days post transduction the binding to recombinant IL13Rα2 and CAR surface expression was assessed by flow cytometry.

## Human T cell engineering, enrichment and expansion

Peripheral blood mononuclear cells (PBMCs) were isolated from human buffy coats obtained from anonymized healthy blood donors. No ethical permit was required as samples were anonymized. PBMCs were activated using either an activation protocol with OKT-3 (100 ng/ml, BioLegend) and IL-2 (100 IU/ml, Proleukin, Novartis) for 3 days at a concentration of $2 \times 10^6$ cells per ml or for 2 days using TransAct reagent (Miltenyi Biotec) according to manufacturer's protocol. Following activation T cells ($1 \times 10^6$ cells) were re-suspended in 20 μl concentrated lentivirus together with 10 mg/ml protamine sulfate (Sigma-Aldrich) or 5 μg/ml polybrene (Sigma-Aldrich) and IL-2 (100 IU) and incubated for 4hrs at 37 °C. T cells were transduced in a similar manner the day after and re-suspended in culturing medium containing 50IU/ml IL-2. After transduction (5–7 days) CAR-T were either used for an analysis directly or expanded using a rapid expansion protocol previously described[28].

## Proliferation and cytotoxicity assays

**In vitro killing assay of glioblastoma cells.** CAR-T cells were rested 2–3 days after expansion at low dose IL-2 (10 IU/ml) before co-culture with IL13Rα2-positive target cells for all experiments. Target cells expressing firefly-luciferase were harvested and washed in PBS before co-culture with engineered T cells at effector (T cell) to target (tumour cell) ratio 0:1, 0.2:1, 1:1, 5:1 and 25:1 for 3–4 days. Firefly luciferase released was used as a measurement of cytotoxicity and was measured using ONE-Glo Luciferase assay system (Promega Biotech AB, Sweden). The relative tumour cell viability of target cells was calculated by normalizing relative light unit (RLU) from co-cultures against the RLU from target cells without T cells.

**In vitro killing assay of THP-1 cells.** CAR-T cells were rested at low dose IL-2 as described above before being co-cultured with firefly luciferase-expressing, CD44v6-positive THP-1 target cells. Experiment was performed at effector to target cell ratio 0:1, 1:1, 3:1 and 10:1 for 4 days. Firefly luciferase released was used as a measurement of cytotoxicity as described above.

**Proliferation assay.** Expanded and rested T cells were labelled with CellTrace Violet Cell proliferation kit (Thermo Fisher scientific) according to manufacturer's protocol. Labelled T cells were either left unstimulated or co-cultured with target U87-MG (UU) cells (1:1 ratio). Cultures were either left untreated or treated with 10 μM lovastatin (Sigma-Aldrich) to prevent proliferation for 4 days before readout using flow cytometry (BD FACSCantoII, BD FACSDiva 8.0.2, BD Bioscience).

## CAR Imaging

T cells were transduced as described above and cultured for 4–5 days. The CAR was then stained using goat anti-human H + L antibody (Supplementary Table S1), fixed with 2% formaldehyde and cells were spun onto a glass slide (1500 rpm for 5 min) and mounted with fluoromount (Southern Biotechnology, Birmingham, AL, USA). The distribution of the CAR on the T cell surface was then evaluated by 40X confocal microscopy (LSM 700, Zeiss, NY, USA) using Zen Blue 1.1.2.0 or Leica SP8 (Leica Microsystems, Wetzlar, Germany) using Leica Application Suite X 3.0.1.15878. The percentage of T cell surface occupied by CAR molecules was calculated by dividing the area occupied by CAR with the total area of the T cell in ImageJ v1.51.

## Flow cytometry

**IL13Rα2 detection.** The expression level of human IL13Rα2 on cell lines and primary cells was assessed by first staining cells with an unconjugated goat anti-IL13Rα2 antibody and after washing staining with a donkey anti-goat IgG conjugated antibody (Supplementary Table S1).

**CD44v6 detection.** The expression level of human CD44v6 on THP-1 cells was assessed by staining with conjugated mouse anti-CD44v6 antibody (Supplementary Table S1).

**scFv binding assay.** scFv binding to target cells was assessed by incubating $1 \times 10^5$ cells with 100 ng FLAG-tagged scFv for 20 min in RT. After washing the cells were stained using a PE-conjugated anti-FLAG antibody for 20 min in RT. Readout was done using CytoFLEX LX Flow cytometer (Beckman Coulter).

**CAR-binding to recombinant IL13Rα2.** To assess CAR-T cell binding to recombinant IL13Rα2, transduced Jurkat cells were incubated with biotinylated human IL13Rα2 (#10350-H08H-B, Sinobiological) for 30 min. Following washing, cells were incubated with streptavidin-APC (#554067, BD Pharmingen) before flow cytometry analysis.

**CAR-T analysis.** For staining of T cell surface markers, cells were washed with FACS Buffer (PBS with 3 mM EDTA and 0.1% bovine serum albumin) and subsequently stained with fluorochrome-conjugated antibodies (Supplementary Table S1). For Annexin-V staining, already surface marker-stained cells were washed with Annexin-V Binding Buffer (BioLegend) and stained with the Annexin-V antibody diluted in binding buffer. For staining of intracellular proteins, cells were washed with ice-cold NaCl (0.9 M) before staining with surface markers. Thereafter the cells were fixed with 2% formaldehyde for 10 min at RT before permeabilizing using ice-cold methanol for 10 min on ice and stained with intracellular markers (Supplementary Table S1) or the True-Nuclear™ Transcription Factor Buffer Set (BioLegend) according to manufacturer's protocol.

CAR-T cell proliferation was assessed by staining cells with Cell-Trace™ Violet Cell Proliferation Kit (Invitrogen, #C34571) according to manufacturer's instructions before staining surface markers.

**CAR expression analysis.** Surface CAR was stained as other T cell surface markers as described above (Antibodies in Supplementary Table S1). When assessing total CAR expression (extracellular and intracellular) the surface CAR was stained first stained and cells washed. Cells were then fixed with 2% formaldehyde for 10 min in RT (In the dark). Subsequently the cells were fixed and permeabilized using True-Nuclear™ Transcription Factor Buffer Set (BioLegend) according to manufacturer's protocol. Intracellular CAR was then stained.

Samples were run in FACSCanto II (BD BioSciences) using BD FACSDiva 8.0.2, AriaIII (BD BioSciences) using BD FACSDiva v9.0.1 or CytoFLEX LX (Beckman Coulter) using CytExpert 2.4. Samples were analysed using FlowJo_v10.8.1 (FlowJo LLC).

## Cytokine detection

Human IFN-γ and IL-2 secreted from either unstimulated CAR-T cells or CAR-T cells co-cultured together with target cells was detected using ELISA. The supernatant from the experimental duplicates was pooled. Before ELISA the supernatant from unstimulated assays were diluted 1:2 or 1:4 and from stimulated cells 1:5. ELISA was then performed in experimental duplicates according to manufacturer's protocol (Mabtech AB, Sweden).

## Gene expression analysis

Mock-T, CAR[B]-T and CAR[E]-T were generated from 4 healthy donors. Transduced T cells were isolated by sorting based on the expression of GFP and RNA extracted using RNeasy Mini Kit (QIAGEN, Hilden, Germany) according to manufacturer's protocol. Gene expression was assessed using the nCounter CAR-T Characterization panel (NanoString Technologies) and the assay was performed at the KIGene Core Facility (Karolinska Institute, Sweden) according to the manufacturers protocol. Gene expression count matrices went through an internal QC process, and were normalized with a set of automatically refined housekeeping genes determined using the default nCounter advanced analysis module with the nSolver4.0 software (NanoString Technologies). Differential gene expression between CAR constructs and the top 20 active signalling pathways was performed using the Fast/Approximate negative binomial model, the adjusted p-values were calculated using Benjamini-Yekutieli method (FDR $p < 0.05$) in nSolver4.0. In addition normalized gene expression data from nSolver were also imported to Partek Flow gene expression analysis software (Partek Inc.) for further analysis (Partek flow 10.0.22.0828). To determine the differentially expressed genes (DEG) the data was first normalized using TMM method and differentially expressed genes were then determined using DESeq2. Genes with fold change less than −1.49 or more than 1.49 and FDR $p < 0.05$ (Supplementary Data file 1 were selected to make a heatmap of the DEG between CAR[B]-T vs CAR[E]-T + Mock-T cells. Activated and inhibited transcription factor pathways were investigated using TfactS (de Duve Institute, Brussels, Belgium) tfacts.org. The significantly up/downregulated DEG between CAR[B]-Ts compared to CAR[E]-Ts + Mock-Ts were used as input (Supplementary Table S2).

## Animal study

Human U-343MG cells stably expressing firefly luciferase (Fluc) were implanted orthotopically ($1 \times 10^5$ cells in 2 μl DPBS per mouse) into the brain of 8-week old female athymic nude mice (Janvier, Rj:ATHYM-Foxn1nu/nu). Tumours were implanted 1 mm anterior and 1.5 mm right from bregma at 2.7 mm depth using a Hamilton syringe and stereotactic equipment (AgnThos AB). Mice were treated with $2 \times 10^6$ Mock-T ($n = 9$), CAR[C]-T ($n = 10$), CAR[D]-T ($n = 10$) or CAR[E]-T ($n = 10$) in 5 μl DPBS 7 days post tumour implantation. The data was pooled from two individual experiments. Tumour growth was monitored by measuring luminescence by in vivo Imaging System (IVIS) imaging. Mice were anaesthetized and injected intraperitoneally (i.p.) with D-luciferin (1.5 mg per mouse in 100 μl) (Sigma, #115144-35-9) before imaging with NightOWL's IVIS (Berthold Technologies). Analysis of IVIS images was performed using the IndiGo software. Mice were euthanized by cervical dislocation when symptoms were developed according to approved ethics guidelines. Experimental and control animals were co-housed. The mice were housed in a barrier facility at an average temperature of 23 °C and humidity of 45-65%. The dark/light cycle was fixed to 12 h. The Northern Stockholm and Uppsala Research Animal Ethics Committee have approved all the animal studies (N164/15, 5.8.18-19434-2019 and 5.8.18-13414.2020).

## Statistical analysis

Statistical analysis for all experiments was performed in GraphPad Prism 9.01 (GraphPad, La Jolla, CA, USA). Statistical comparison between the means of two experimental groups was performed using an unpaired two-tailed t-test. If comparing several experimental groups with one experimental reference group a one-way ANOVA was performed with Dunnette's correction for multiple comparison. If comparing more than two experimental groups a one-way ANOVA with Tukey's correction for multiple comparison was used. A p-value of less or equal to 0.05 was considered statistically significant. The p-values were represented as; *$p \le 0.05$, **$p \le 0.01$, ***$p \le 0.001$, ****$p \le 0.0001$.

## Reporting summary

Further information on research design is available in the Nature Portfolio Reporting Summary linked to this article.

## Data availability

The gene expression data generated in this study has been deposited in the Mendeley Data database under accession code: https://doi.org/10.17632/k4r8ynxyz9.3[29]. The access to other pre-processed data files (e.g. flow cytometry, images, SPR etc.) can be obtained by contacting the corresponding authors due to large data sets. Sequence for extracellular domain of IL13Rα2: Uniprot Q14627 aa 29-501 Source data are provided with this paper.

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

## Acknowledgements

The authors would like to thank Vendela Parrow, Annette Roos, Esmeralda Woestenenk and Kristian Sandberg at SciLifeLab for assistance in coordinating the scFv screening process. The authors would also like to thank Minttu-Maria Martikainen for assistance with animal experiments. This work was supported by research grants from the Swedish Research Council (2019-01326), the Swedish Cancer Society (19 0184Pj), the Swedish Childhood Cancer Fund (PR2020-0167), the Sjöberg Foundation (2020-01-07-06), and SciLifeLab (DP_ME_107). M.R. was supported by postdoctoral grant from the Swedish Childhood Cancer Fund (TJ 2019-0014).

## Author contributions

T.S. designed research, performed experiments, analysed and interpreted data, and was involved in preparing the manuscript. G.S., P.M.T., X.Z., J.T., Y.A., and C.H. performed experiments and analysed data. M.N. provided material and sequences. A.D. and H.P. interpreted data. M.R. designed research, analysed and interpreted data and was involved in preparing the manuscript. D.Y and M.E. supervised and designed research, interpreted data, and were involved in preparing the manuscript.

## Funding

## Competing interests

D.Y. and M.E. are co-founders of Elicera Therapeutics, which has submitted a patent application on the described IL13Ra2-directed scFv sequences. M.N. is co-founder of Akiram Therapeutics, which has submitted a patent application on CD44v6-targeted antibodies. The remaining authors declare no competing interests.
