## [Peer Review File · Nature Communications]

Complementarity-determining region clustering may cause CAR-T cell dysfunctionREVIEWER COMMENTS

Reviewer #1 (Remarks to the Author):

In this manuscript by Saren et al., the authors identified and evaluate several IL13Ra2-targeting CAR constructs with varying scFvs and their effector function in vitro and in vivo. They identified differential activity of these scFv-based CARs, and that one scFv (CAR(B)) had greater antigen-independent tonic signaling. Importantly, changes in sequence of CAR(B) in the CDR reduced clustering, tonic signaling by lower IFN γ production and activation/exhaustion markers. However, several key studies are lacking in this well-executed and displayed work. Critical data on these mutant CAR(B) constructs, including cytotoxicity, T cell expansion/proliferation, and in vivo therapy are missing. These studies would be required to confirm that these changes in tonic signaling are not simply due to a lack of CAR T cell activity/targeting of IL13Ra2. Further, it is unclear whether this CDR-driven clustering phenomenon extends beyond IL13Ra2, and if not, this would limit the broad applicability of these findings to the CAR field. Nonetheless, the following could be address to improve the impact of this study:

1. CAR expression on the surface of T cells should be shown for all 5 CARs associated with Figure 1. Is there a difference in CAR surface expression in these different CARs?
2. While the % binding of the 5 CARs to U87 was fairly similar, it is clear that MFI is lowest likely in E, suggesting that affinity measures to hIL13Ra2 and to cells expressing IL13Ra2 may be different. Please explain in the context of tonic signaling. If affinity is different between these 5 CARs, it is difficult to interpret the differences in on-target activity as well as tonic signaling.
3. While the authors show scFv(B) as a soluble form being able to bind to IL13Ra2 molecules on tumor cells, they do not show antigen-specific binding in the CAR form; the authors should stain CAR(B) T cells with recombinant soluble IL13Ra2, and compared with CAR(E). This would also confirm that lack of activity of CAR(B) is due to tonic signaling, and not just due to lack of binding to target.
4. Since there seems to be a lack of IFN γ and T cell activation of the 5 scFvs against antigen-negative tumors cells, but significant activation/exhaustion markers shown in Figure 2 in resting CAR(B) T cells, it would be prudent to justify the discrepancy and explain in more detail the differences between these two experimental conditions.
5. Critical data is missing here in the mutant CAR(B) constructs, specifically T cell cytotoxicity, T cell expansion/proliferation, and in vivo therapy activity in comparison to WT CAR(B) and CAR(E).
6. Additionally, affinity measures of these variants should be performed to compare to CAR(B) and specifically confirm that all functional readouts in Figure 4 is not due to loss of affinity of the CAR variants. Otherwise, the diminished cell size, IFN γ production, and activation/exhaustion markers may be due to disruption/reduction of antigen binding and/or loss of CAR T cell activity.

Additional comments:

1. For all data, please include timepoints associated with analysis for each.
2. Flux in Figure 1R should be presented in log₁₀ scale, and consider placing survival after flux imaging and quantification.
3. IL13Ra2 expression flow plots (associated with MFI in Figure S2 should be shown comparing U87 and U343 lines associated with Figure 1.

Reviewer #2 (Remarks to the Author):

In this manuscript, Sarén et al. report the generation of CAR-T cells derived from five different scFv that only differed in the CDR loop sequences. They demonstrate that CDR sequences have an impact on CAR expression and can drive tonic signaling, which may result in impaired anti-tumor effects. The manuscript is well-written, and the experiments are well-designed, including the necessary controls. Overall, the quality of this study is high, and the conclusions are largely supported by the data. However, while the findings in this work are interesting, there are some significant weaknesses that raise many questions. The authors need to address the following

concerns in order to enhance the quality of this study:

This work addresses an immunologically relevant issue to understand the behavior of different CAR constructs. However, there is enough evidence in the field showing that the combination of certain scFv and costimulatory domains can drive tonic signaling. The main question to resolve is how can we design CARs whose configuration do not induce tonic signaling. Is there a specific amino acid combination responsible for CAR oligomerization? Can we predict CAR oligomerization based on computational modelling techniques to be used in silico? Can we mutate the CDR to prevent CAR-mediated tonic signaling while improving CAR-mediated killing? Would the combination of the scFv B with a different costimulatory domain (i.e. CD28) result in less tonic signaling? The lack of solutions to the issues highlighted in this manuscript render their results incomplete.

Tonic signaling is dependent on the levels of CAR expression. Flow cytometry plots showing CAR expression should be included in Figure 1 for scFv A-E and in Figure 4 for all mutants. Also, the authors state in the discussion that CAR expression on CAR(B) T cells is reduced over time. These results should be included in the main manuscript.

A functional characterization of the CAR(B) mutants upon antigen recognition should be included.

Why did the authors choose to characterize tonic signaling only on CAR(B) T cells? Do the other scFv also induce tonic signaling? Rescuing CAR(B)-T cell efficacy by mutating the CDR may be challenging, but could this strategy be used to improve the in-vivo antitumor efficacy of CAR(C) and CAR(D), which showed promising in vitro efficacy?

Finally, the authors state that they have identified a candidate CAR-T that warrant clinical translation for the treatment of recurrent glioblastoma. However, they have only tested the candidate CAR-T cells in one in vivo experiment. The anti-tumor efficacy of CAR(E)-T cells should be tested in an additional animal model.

Minor comments:

- To which treatment group do the animals shown in Figure 1S belong?
- The authors stated: "we created a double mutant, with wild-type H2 and the most common L3 sequence ([B]-H2L3wt)"- What do authors mean by "the most common L3 sequence"? How did they choose mutations in CDR2 of the heavy chain and CDR3 of the light chain?
- The in vivo antitumor efficacy of CAR-T C and D should be moved from supplementary data to Figure 1.

Reviewer #3 (Remarks to the Author):

The authors describe the generation of a set of 5 human IL13Ra2-specific Fc ν molecules that did not cross-react with human IL13Ra1. They show that CAR constructs made with these Fc ν s have variable levels of activation in response to cell lines expressing IL13Ra2 in vitro. T cells with their top candidate, CAR[E], showed good activation against various cell lines expressing IL13Ra2 at differing levels and efficient activity in an in vivo model.

During this characterisation they found that some of the candidate Fc ν s with high affinity to target (CAR[A] and CAR[B]) have a high degree of tonic signalling and carefully characterised CAR[B] against their best candidate CAR[E]. They find upregulation of exhaustion markers and dysfunction in CAR[B] in the absence of ligand, whilst the expression profile of CAR[E] was very similar to mock transduced cells.

They show that dysfunction in CAR[E] is likely due to CAR clustering and tonic signalling. The authors looked at a number of activation markers after exposing CAR[B]-T and CAR[E]-T cells to antigen expressing cells and found that, consistent with early cytokine secretion data, only CAR[E]-T cells were responsive.

Finally, they performed alanine mutations of residues in the CDR loops of CAR[B] that differed between CAR[B] and CAR[E]. They find that mutation of some of these residues prevents aggregation of the construct and rescues the tonic signalling. Finally, they show that tonic signalling is mediated by the cytoplasmic tail in the aggregated forms of the receptors by generating tail-less "decoy" receptors, showing tail-less CAR[B] does not induce tonic signalling.

In general the quality of the work is very good, but I am left wondering what the specific message of the paper is. It seems to have two separate parts, the first being a description of a candidate CAR for glioblastoma treatment and the second an in depth analysis of why some of the candidate CAR constructs failed to work. Perhaps some better integration of the two parts in the discussion would help me as a reader understand whether this study is primarily describing a CAR construct that will be taken further into clinical trials, or whether it is meant to be instructive for others developing CAR therapies and describing mechanisms that lead to tonic signalling in constructs.

Some specific comments:

- Figure 2B: It doesn't look like PC3 is needed to separate the conditions (unless there was a difference in PC3 between Mock and CAR[E]-T). The figure as plotted also makes it impossible to distinguish differences in PC3. The authors could consider plotting the data on a 2D PC1 vs PC2 with the size of points scaled to values of PC3, or have 3 plots of PC1 vs PC2, PC1 vs PC3 and PC2 vs PC3 or perhaps just leaving PC3 out and plotting PC1 vs PC2 and mentioning that PC3 did not aid in separating categories.

- Figure 4: To tie together the concept that aggregation induces signalling through the cytoplasmic tail, data showing that CAR[B]d aggregates like CAR[B] (like the data shown in panel E and quantified in panel F) is required. This would also confirm that aggregation is not mediated by interactions between signalling molecules associated with the cytoplasmic tails of receptors activated by some potential conformational mechanism.

- As the authors note in the discussion, it is well known that CDR loop sequences can induce aggregation, and that aggregation of CARs induces tonic signalling. Most published work has focussed on the contribution of the framework and hinge regions of CARs to aggregation and tonic signalling, and as such this is one of the few to follow up candidate CARs with high tonic signalling and show that the CDR loops in the scFv are responsible. Most others would assume this is the case and discard the candidate. The bigger question is whether promising scFv candidates can be rescued. The authors used alanine mutants or substitutions of sequences ([B]-H3L2wt) to show that tonic signalling can be reversed, but determining to what degree this has affected binding to antigen or how effectively CAR-T cells with these constructs can be activated by cells expressing the target antigen would be useful.

Response to specific questions to each reviewer

Reviewer #1 (Remarks to the Author):

In this manuscript by Saren et al., the authors identified and evaluate several IL13Ra2-targeting CAR constructs with varying scFvs and their effector function *in vitro* and *in vivo*. They identified differential activity of these scFv-based CARs, and that one scFv (CAR(B)) had greater antigen-independent tonic signaling. Importantly, changes in sequence of CAR(B) in the CDR reduced clustering, tonic signaling by lower IFN γ production and activation/exhaustion markers. However, several key studies are lacking in this well-executed and displayed work. Critical data on these mutant CAR(B) constructs, including cytotoxicity, T cell expansion/proliferation, and *in vivo* therapy are missing. These studies would be required to confirm that these changes in tonic signaling are not simply due to a lack of CAR T cell activity/targeting of IL13Ra2. Further, it is unclear whether this CDR-driven clustering phenomenon extends beyond IL13Ra2, and if not, this would limit the broad applicability of these findings to the CAR field. Nonetheless, the following could be address to improve the impact of this study:

We greatly appreciate the reviewer's positive feedback. We agree with the reviewer that we did not show that CDR-driven clustering and subsequent antigen-independent tonic signaling is a general phenomenon extending beyond IL13Ra2-targeting CARs. Therefore, in the updated version of the manuscript we have demonstrated CDR-mediated clustering for an additional IL13Ra2-targeting CAR, namely CAR[A]-T cells in Supplementary Figure S8. Furthermore, we show CDR-mediated clustering can also be observed in CARs targeting CD44v6 in Supplementary Figures S9 and S10. We thereby strengthen that our findings have a broader applicability in the CAR-T cell field.

1. CAR expression on the surface of T cells should be shown for all 5 CARs associated with Figure 1. Is there a difference in CAR surface expression in these different CARs?

We thank the reviewer for pointing this important aspect out. There is a difference in CAR surface expression between the 5 CARs and in the previous version of the manuscript we only showed it for CAR[B] and CAR[E] in Supplementary Figure S5B. In the revised manuscript, we have added this information for CAR[A], CAR[B] and CAR[E] into the main Figure 2 and for all 5 CARs in Supplementary Figure S6A. We believe that the difference between surface expression between the 5 CARs is due to CAR stability with CAR[A] and CAR[B] being unstable leading to reduce surface expression and lower functionality. This is shown in the revised manuscript in the Results section (lines 119-125).

2. While the % binding of the 5 CARs to U87 was fairly similar, it is clear that MFI is lowest likely in E, suggesting that affinity measures to hIL13Ra2 and to cells expressing IL13Ra2 may be different. Please explain in the context of tonic signaling. If affinity is different between these 5 CARs, it is difficult to interpret the differences in on-target activity as well as tonic signaling.

We thank the reviewer for allowing us to explain this further. As the tonic signaling evaluated in this manuscript is an antigen-independent phenomenon (performed on resting CAR-T cells that have not seen the cognate CAR target antigen) the affinity of the CARs to the target antigen will most likely not influence antigen-independent tonic signaling. We agree that the difference in affinity between the 5 CARs could have an impact in terms of on-target activity. However, published literature on this have shown that high affinity CARs have normally better function *in vitro* than low affinity CARs (PMID: 26330166, PMID: 29085043). In our data, scFv[A]-scFv[C] have the highest affinity to IL13Ra2 (Figure 1B) which also reflects the binding of CAR[A]-[C] to IL13Ra2 (Supplementary Figure S2A). As expected, CAR[C]-T is highly functional, however, CAR[A]-T and CAR[B]-T display poor

functionality despite their high affinity. Furthermore, although CAR[C] has higher affinity than CAR[D] and CAR[E], CAR[E]-T performs better. As our data does not follow the common understanding of the field, we believe that affinity and target binding is not the major contributor to the results we observe but instead the CDR-mediated CAR clustering which results in antigen-independent activation and subsequent tonic signaling and CAR-T cell exhaustion.

3. While the authors show scFv(B) as a soluble form being able to bind to IL13Ra2 molecules on tumor cells, they do not show antigen-specific binding in the CAR form; the authors should stain CAR(B) T cells with recombinant soluble IL13Ra2, and compared with CAR(E). This would also confirm that lack of activity of CAR(B) is due to tonic signaling, and not just due to lack of binding to target.

We thank the reviewer for this comment. We have now performed the suggested experiment using recombinant soluble IL13Ra2 to show the binding of the 5 CARs to the receptor. The binding data for the 5 CARs is shown in Figure S2A in the revised manuscript. All 5 CARs bind to recombinant IL13Ra2 (Supplementary Figure S2A) at levels matching the affinity of the corresponding scFv (Figure 1B).

New experiments performed for this revision further suggest that the lack of activity of CAR[B] is a consequence of tonic signaling. In the updated version of the manuscript, we have evaluated the response of CAR[B] and CAR[B] mutants to antigen stimulation (Supplementary Figure S7). CAR[B] with mutation in CDR3 of the light chain, named [B]-L3, display reduced tonic signaling and subsequently respond better to antigen-stimulation compared to CAR[B], while both constructs bind similarly well to IL13Ra2.

4. Since there seems to be a lack of IFN γ and T cell activation of the 5 scFvs against antigen-negative tumors cells, but significant activation/exhaustion markers shown in Figure 2 in resting CAR(B) T cells, it would be prudent to justify the discrepancy and explain in more detail the differences between these two experimental conditions.

We appreciate that the reviewer pointed this out and gave us the opportunity to explain this better. IFN- γ secretion from CAR-T cells into the co-culture supernatant of CAR-T cells together with antigen-negative tumor cells (Mel526) was evaluated by ELISA (Supplementary Figure S2K). In Figure 2 (Figure 3 in the revised manuscript) the gene expression was assessed. Gene expression analysis is more sensitive than ELISA and thus explain why although activity might be evident at the gene expression level no IFN- γ can be detected by ELISA.

We would also like to take the opportunity to clarify the difference between the experimental settings used to obtain the different samples used for IFN- γ detection in the manuscript. CAR[A]-T cells and CAR[B]-T cells secrete IFN- γ into the culture supernatant even without antigen-stimulation (Fig. 2I). Despite this, for CAR[A]-T cells and CAR[B]-T cells, no IFN- γ was detected upon co-culture with antigen-negative (Mel526) cells (Fig. S2K) and very low levels of IFN- γ were detected after antigen stimulation (Fig. 1G-H; Fig. S2F-G). This can be explained as the assays were performed at different time points after T cell transduction. The co-culture experiment was set up at a later time point after T cell transduction and rapid expansion, and due to prolonged exposure to tonic signaling the ability to secrete IFN- γ was reduced. We have clarified this in the revised manuscript by adding at what time point after T cell transduction the experiments were performed in all Figure legends.

5. Critical data is missing here in the mutant CAR(B) constructs, specifically T cell cytotoxicity, T cell expansion/proliferation, and in vivo therapy activity in comparison to WT CAR(B) and CAR(E).

We thank the reviewer for this comment and we have performed new experiments to address this. We have extended the evaluation of antigen-independent tonic signaling of CAR[B] mutants in Figure 4. As suggested, we have also evaluated the response of the CAR[B] mutants to antigen stimulation (Supplementary Figure S7C-J). However, we would like to highlight that CAR[B] was mutated to confirm that the CDRs were responsible for CAR clustering and subsequent antigen-independent tonic signaling. Thus, we did not intend to rescue CAR[B] with kept affinity for the target antigen through mutation of the CDR loops as this certainly may impair binding and specificity to the target antigen. As expected, the non-tonically signaling [B]-H2L3wt did no longer bind to the target antigen. [B]-L3 on the other hand, which displayed partially reduced tonic signaling, showed slightly better response upon antigen stimulation compared to CAR[B].

6. Additionally, affinity measures of these variants should be performed to compare to CAR(B) and specifically confirm that all functional readouts in Figure 4 is not due to loss of affinity of the CAR variants. Otherwise, the diminished cell size, IFN γ production, and activation/exhaustion markers may be due to disruption/reduction of antigen binding and/or loss of CAR T cell activity.

We apologize for not being clear enough in our description of Figure 4. All studies with the CAR[B] mutants displayed in Figure 4 were performed without antigen-stimulation to show the effect of the different mutations on antigen-independent tonic signaling. We have now emphasized this both in the manuscript text and Figure 4 Legend regarding the experimental setting.

We thank the reviewer for pointing out the potential change in the binding for these mutants and have now evaluated the binding of the CAR[B] mutants to recombinant human IL13R α 2. It is presented in Supplementary Figure S7B.

Additional comments:

1. For all data, please include timepoints associated with analysis for each.

We thank the reviewer for pointing this out. We have now addressed this issue and added all time points at which the assays were performed in the figure legends.

2. Flux in Figure 1R should be presented in log₁₀ scale, and consider placing survival after flux imaging and quantification.

We thank the reviewer for addressing this. As both Reviewer #3 and the editor think that we have 2 stories in our original manuscript, we have now placed the *in vivo* data in Supplementary Figure S3 and as suggested present the tumor growth in log₁₀ scale.

3. IL13R α 2 expression flow plots (associated with MFI in Figure S2 should be shown comparing U87 and U343 lines associated with Figure 1.

A histogram displaying the IL13R α 2-expression in the glioblastoma cell lines have now been added in Supplementary Figure S2C to better show the expression level of IL13R α 2 in the different cell lines.

Reviewer #2 (Remarks to the Author):

In this manuscript, Sarén et al. report the generation of CAR-T cells derived from five different scFv that only differed in the CDR loop sequences. They demonstrate that CDR sequences have an impact on CAR expression and can drive tonic signaling, which may result in impaired anti-tumor effects. The manuscript is well-written, and the experiments are well-designed, including the necessary controls. Overall, the quality of this study is high, and the conclusions are largely supported by the data. However, while the findings in this work are interesting, there are some significant weaknesses that raise many questions. The authors need to address the following concerns in order to enhance the quality of this study:

This work addresses an immunologically relevant issue to understand the behavior of different CAR constructs. However, there is enough evidence in the field showing that the combination of certain scFv and costimulatory domains can drive tonic signaling. The main question to resolve is how can we design CARs whose configuration do not induce tonic signaling. Is there a specific amino acid combination responsible for CAR oligomerization? Can we predict CAR oligomerization based on computational modelling techniques to be used *in silico*? Can we mutate the CDR to prevent CAR-mediated tonic signaling while improving CAR-mediated killing? Would the combination of the scFv B with a different costimulatory domain (i.e. CD28) result in less tonic signaling? The lack of solutions to the issues highlighted in this manuscript render their results incomplete.

We greatly appreciate the positive feedback from the reviewer. We agree that one of the major tasks in the field of antibody, scFv and CAR engineering is to predict and be able to avoid oligomerization and thereby be able to design a CAR devoid of clustering and subsequent tonic signaling. We have unfortunately not been able to resolve this issue in the current manuscript. The aim with our manuscript is to report on a new phenomenon: that CDR-mediated clustering, without any antigen stimulation, can cause tonic signal in CAR-T cells. So far, only the framework region of the scFv has been described associated with tonic signaling. In addition, to help scientists in selecting CARs to avoid tonic signaling, we also proposed that the activity of resting (not stimulated by the cognate antigen) CAR-T cells, *e.g.*, enlarged cell size and background IFN-gamma secretion could be used as an easy method to exclude CARs that are prone to tonic signaling. This is important, as we suggest in the manuscript that CAR clustering and tonic signaling cannot be predicted at the scFv-level but must be evaluated at the CAR-T cell level. For example, the melting temperature (thermal stability) of scFv[B] is similar to that of the FMC63 clone used for construction of clinically approved CD19-directed CAR-T cells. Yet CAR[B]-T cells are prone to tonic signaling while, CD19 CAR-T cells are not. We have further strengthened the importance of our findings to the CAR-T cell field by showing that CDR-mediated tonic signaling is a general phenomenon seen in CARs against another target than IL13R α 2, in this case against CD44v6 (Supplementary Figures S9 and S10).

We also believe that changing the co-stimulatory domain from 4-1BB to *e.g.*, CD28 would not rescue the tonic signaling as 4-1BB has been shown to result in lower tonic signaling compared to CD28 (doi:10.1038/nm.3838, PMID 25939063). Additionally, our data suggests the tonic signaling was caused by CDR-mediated scFv clustering, which further leads to signal transduction via the downstream co-stimulatory domain leading to non-responsive CAR-T cells.

Tonic signaling is dependent on the levels of CAR expression. Flow cytometry plots showing CAR expression should be included in Figure 1 for scFv A-E and in Figure 4 for all mutants. Also, the

authors state in the discussion that CAR expression on CAR(B) T cells is reduced over time. These results should be included in the main manuscript.

We agree with the reviewer that the level of tonic signaling can vary with CAR expression and that for a CAR prone to clustering, higher CAR expression would likely lead to higher tonic signaling. For a CAR not prone to clustering, such as CAR[E], the expression level of CARs on the surface of T cells is retained over time while for a CAR prone to clustering, such as CAR[B], clustering will lead to aggregation and reduced level of CAR on the surface of the T cells over time. We have now added the requested data of expression level to the revised version of the manuscript, as main Figure 2B for CAR[A], CAR[B] and CAR[E], Supplementary Figure S6A for CAR[A]-[E], and Supplementary Figure S7A for CAR[B] mutants. Further the reduced CAR expression over time has been added to the main manuscript in Figure 2C-E.

A functional characterization of the CAR(B) mutants upon antigen recognition should be included.

We thank the reviewer for pointing this out and have now performed a more detailed characterization of the CAR[B] mutants both without antigen stimulation and with antigen stimulation. These data can now be found in Figure 4 and Supplementary Figure S7. However, we would also like to highlight that the aim of creating the CAR[B] mutants was to confirm that the CDR loops caused CAR clustering and subsequent antigen-independent tonic signaling and determine which CDR loops were responsible for clustering in CAR[B]. In the revised manuscript we have performed mutation of another IL13R α 2-directed CAR that was identified in our screen, CAR[A] (Supplementary Figure S8). In general, the data obtained from mutating the CDRs of CAR[A] corroborate the data already presented for CAR[B] in the first version of the manuscript. We have also assessed the binding of all mutants to recombinant IL13R α 2. In one case, when mutating CDR3 of the light chain for CAR[B], named [B]-L3, we observed reduced tonic signaling with maintained binding to IL13R α 2. This was however not the case when simultaneously mutating CDR3 of the light chain and CDR2 of the heavy chain for CAR[B], named [B]-H2L3wt, where tonic signaling was completely abolished but the binding to IL13R α 2 was at the same time lost. When mutating CAR[A], we found that it was mainly CDR3 of the heavy chain that was mediating clustering. When it was mutated, clustering was abolished but binding to IL13R α 2 was at the same time lost.

Why did the authors choose to characterize tonic signaling only on CAR(B) T cells? Do the other scFv also induce tonic signaling? Rescuing CAR(B)-T cell efficacy by mutating the CDR may be challenging, but could this strategy be used to improve the in-vivo antitumor efficacy of CAR(C) and CAR(D), which showed promising in vitro efficacy?

We thank the reviewer for bringing this up. The reasons why we chose to characterize CAR[B]-T cells further was that it was the most tonically signaling CAR among the 5 CARs and had the lowest cytotoxic potential against target cells. We agree with the reviewer that CAR[B]-T might be the most difficult CAR to rescue. However, we would like to emphasize that the intention with the mutation analysis was not to rescue the CAR-T functionality but to determine if the amino acids in the CDR regions were responsible for CAR clustering. This is also the reason we replaced corresponding CDRs using the nonpolar amino acid alanine, to minimize the side chain interaction and thus reduce the likelihood for clustering. Indeed, the binding activity towards IL13R α 2 was affected by introducing mutations as presented in Supplementary Figure S7.

In the revised manuscript we also mutated CAR[A] to show that the CDRs were also causing tonic signaling in CAR[A]-T cells (Supplementary Figure S8). Clustering could be avoided by mutating certain amino acids but at the cost of compromising binding capacity. To further strengthen our

findings, we also evaluated if CDR-mediated clustering was a general phenomenon existing beyond IL13R α 2. Importantly we found that CDR-mediated CAR-clustering of CD44v6-targeting CARs could induce antigen-independent tonic signaling (Supplementary Figures S9 and S10) showing generalizability of our findings.

We have indications that CAR[C], CAR[D] and CAR[E]-T cells display a low degree of tonic signaling as they have a somewhat larger cell size and secrete low levels of IFN- γ without antigen-stimulation in comparison to Mock-T control. We believe that this level of tonic signaling might even be beneficial to lowering the activation threshold of these constructs without inducing dysfunction. A recent study has shown that a low level of tonic 4-1BB signaling is even beneficial for CAR T cells (doi:10.1038/s41591-021-01326-5, PMID: 33888899).

Finally, we got the comment from the editor that the paper had two parallel stories, tonic signaling which we consider the most important one for this manuscript and the therapeutic efficacy against glioblastoma *in vivo* with further clinical translation. Therefore, the *in vivo* anti-tumor efficacy part has been reduced in the revised manuscript.

Finally, the authors state that they have identified a candidate CAR-T that warrant clinical translation for the treatment of recurrent glioblastoma. However, they have only tested the candidate CAR-T cells in one *in vivo* experiment. The anti-tumor efficacy of CAR(E)-T cells should be tested in an additional animal model.

We agree that if the main focus of the paper would have been to promote CAR[E]-T cells for clinical translation an additional model would certainly be needed. However, following the suggestion of the editor and focusing our paper, we have formatted the manuscript in a different way, now mainly focusing on the tonic signaling part, while taken out the therapeutic part into Supplementary Figure S3 to make the story coherent. By downscaling the therapeutic part, the editor did not suggest us to add a second animal model (please read the editor's comment), but instead focus to the main story line. We hope that this is acceptable.

Minor comments:

- To which treatment group do the animals shown in Figure 1S belong?

We have put the therapeutic data into Supplementary Figure S3 in the revised manuscript and made sure that it is clear which images belongs to which treatment group.

- The authors stated: "we created a double mutant, with wild-type H2 and the most common L3 sequence ([B]-H2L3wt)"- What do authors mean by "the most common L3 sequence"? How did they choose mutations in CDR2 of the heavy chain and CDR3 of the light chain?

We apologize for not explaining this properly and thank the reviewer for bringing this up. We used a human synthetic fragment library in our phage display library to obtain the single chain variable fragments. The particular CDR-L3 sequence we used in the mutant is the most frequently occurring CDR-L3 sequence within functional antibodies of the human IGKV1-39 germline that our library was based upon. For CDR-H2 the wild type H2 sequence is from the human IGHV3-23 germline, which the library was based on for the heavy chain, was used. We have now added additional sentences both in the result section (line 243) and materials and methods section (lines 392-393) of the revised manuscript to clarify this.

- The *in vivo* antitumor efficacy of CAR-T C and D should be moved from supplementary data to Figure 1.

To get the main message out, which is that certain CDR sequences or combination of CDR sequences can lead to CAR clustering, tonic signaling and CAR-T dysfunction we have instead moved out the therapeutic *in vivo* data from the main Figure into Supplementary Figure S3, in order to make the revised manuscript focused on the technical aspect and the role of the CDRs in CAR clustering.

Reviewer #3 (Remarks to the Author):

The authors describe the generation of a set of 5 human IL13R α 2-specific Fc γ molecules that did not cross-react with human IL13R α 1. They show that CAR constructs made with these Fc γ s have variable levels of activation in response to cell lines expressing IL13R α 2 *in vitro*. T cells with their top candidate, CAR[E], showed good activation against various cell lines expressing IL13R α 2 at differing levels and efficient activity in an *in vivo* model.

During this characterisation they found that some of the candidate Fc γ s with high affinity to target (CAR[A] and CAR[B]) have a high degree of tonic signalling and carefully characterised CAR[B] against their best candidate CAR[E]. They find upregulation of exhaustion markers and dysfunction in CAR[B] in the absence of ligand, whilst the expression profile of CAR[E] was very similar to mock transduced cells.

They show that dysfunction in CAR[E] is likely due to CAR clustering and tonic signalling. The authors looked at a number of activation markers after exposing CAR[B]-T and CAR[E]-T cells to antigen expressing cells and found that, consistent with early cytokine secretion data, only CAR[E]-T cells were responsive.

Finally, they performed alanine mutations of residues in the CDR loops of CAR[B] that differed between CAR[B] and CAR[E]. They find that mutation of some of these residues prevents aggregation of the construct and rescues the tonic signalling. Finally, they show that tonic signalling is mediated by the cytoplasmic tail in the aggregated forms of the receptors by generating tail-less "decoy" receptors, showing tail-less CAR[B] does not induce tonic signalling.

In general the quality of the work is very good, but I am left wondering what the specific message of the paper is. It seems to have two separate parts, the first being a description of a candidate CAR for glioblastoma treatment and the second an *in depth* analysis of why some of the candidate CAR constructs failed to work. Perhaps some better integration of the two parts in the discussion would help me as a reader understand whether this study is primarily describing a CAR construct that will be taken further into clinical trials, or whether it is meant to be instructive for others developing CAR therapies and describing mechanisms that lead to tonic signalling in constructs.

We appreciate the positive feedback from the reviewer. We agree that the paper was written in a way so that it looked like two separate parts, and we got the same comment from another reviewer and the editor. We have therefore now re-written it in a way to make the message of the paper clearer and only focus on the tonic signaling part. In light of this, we have also extended our evaluation of antigen-independent tonic signaling beyond IL13R α 2. In the revised manuscript we show that CDR-mediated tonic signaling is also observed in CARs directed against CD44v6 (Supplementary Figures S9 and S10).

Some specific comments:

- Figure 2B: It doesn't look like PC3 is needed to separate the conditions (unless there was a difference in PC3 between Mock and CAR[E]-T). The figure as plotted also makes it impossible to distinguish differences in PC3. The authors could consider plotting the data on a 2D PC1 vs PC2 with the size of points scaled to values of PC3, or have 3 plots of PC1 vs PC2, PC1 vs PC3 and PC2 vs PC3 or perhaps just leaving PC3 out and plotting PC1 vs PC2 and mentioning that PC3 did not aid in separating categories.

We thank the reviewer for pointing this out and we agree with the comment. We have now changed this figure into a 2D PCA instead (Figure 3B in the revised manuscript).

- Figure 4: To tie together the concept that aggregation induces signalling through the cytoplasmic tail, data showing that CAR[B]d aggregates like CAR[B] (like the data shown in panel E and quantified in panel F) is required. This would also confirm that aggregation is not mediated by interactions between signalling molecules associated with the cytoplasmic tails of receptors activated by some potential conformational mechanism.

We thank the reviewer for this valid point. We have now studied the aggregation of CAR[B]d and compared it to CAR[E]d and seen that CAR[B]d still aggregates. This data has now been added into Supplementary Figure S5B in the revised manuscript.

- As the authors note in the discussion, it is well known that CDR loop sequences can induce aggregation, and that aggregation of CARs induces tonic signalling. Most published work has focussed on the contribution of the framework and hinge regions of CARs to aggregation and tonic signalling, and as such this is one of the few to follow up candidate CARs with high tonic signalling and show that the CDR loops in the scFv are responsible. Most others would assume this is the case and discard the candidate. The bigger question is whether promising scFv candidates can be rescued. The authors used alanine mutants or substitutions of sequences ([B]-H3L2wt) to show that tonic signalling can be reversed, but determining to what degree this has affected binding to antigen or how effectively CAR-T cells with these constructs can be activated by cells expressing the target antigen would be useful.

We agree that to rescue promising scFv candidates is one of the biggest challenges in the field of antibody and CAR-T engineering. A universal solution to this would be remarkably important. Unfortunately, we cannot provide such solution with our findings. Importantly, our data clearly suggest that screening of scFvs as recombinant proteins is not sufficient but that the scFvs need to be screened in the CAR format to determine if they cause tonic signaling. In the revised manuscript, we strengthened our findings by proposing to use cell size and autonomous IFN-gamma secretion of unstimulated CAR-T cells as easy method for CAR candidates screening.

We performed alanine substitution with the aim to confirm that the tonic signaling is CDR amino acids dependent, rather than to rescue these scFvs. In addition, we have now performed experiments to determine how alanine mutation will affect binding to IL13R α 2 and thus affect the function of CAR-Ts. New experiment data has been presented in the revised manuscript in Supplementary Figure S7. In one case, when mutating CDR3 of the light chain for CAR[B], named [B]-L3, we observed reduced tonic signaling with maintained binding to IL13R α 2. This was however not the case when simultaneously mutating CDR3 of the light chain and CDR2 of the heavy chain for CAR[B], named

[B]-H2L3wt, where tonic signaling was completely abolished but the binding to IL13R α 2 was at the same time lost. Thus, as expected alanine mutations is not a successful method to recuse scFvs/CARs.

REVIEWERS' COMMENTS

Reviewer #1 (Remarks to the Author):

In the revised manuscript by Saren et al, the authors have addressed several critiques made in the initial review. The overall phenomenon of tonic signaling, exhaustion, and differences in activity are appreciated through modifications to the CDR of scFvs. However, it is still unclear whether this is a general phenomenon that requires solving for future CAR development. It is also noteworthy that several other IL13Ra2-CARs and CD44v6-CARs have been developed that lacked this type of tonic signaling, and did not necessarily require this depth in analysis. Therefore, there is still the question of whether this is a niche story in the development of one or two novel scFv-based CARs.

1. Please explain why excess IL-13 was able to block scFv binding to IL13Ra2 with A-C, but not with D-E? This is a phenomenon that would likely partially explain antigen-dependent differences that may confound the antigen-independent exhaustion from tonic signaling. While these data were included in the initial submission, it was rewritten slightly this round in a way that caught the reviewer's attention, coupled with other revisions performed to address more differences between A-C and D-E. Overall, many differences are apparent with A-C vs D-E, but it is appreciated that the authors demonstrated that mutants in the CDR of the scFvs could reduce tonic signaling and antigen-dependent stimulation of the CAR T cells.

2. In the revised in vivo data (fig S3), CAR-D shows very little efficacy, compared with CAR-E. Yet both, to the extent that they were compared in vitro, showed little tonic signaling and comparable anti-tumor activity (except for lower IL-2 expression, fig. S2h-i, and higher Pd1 expression, S6h). Again, making the reviewer wonder whether tonic signaling is the major aspect contributing to differences between the 5 scFvs.

3. The reviewer appreciates the inclusion of CD44v6 scFvs and clustering phenomenon between two CARs. However, affinities were not measured, and with the variations with in vivo therapy using IL13Ra2-CAR-D and CAR-E, which both showed a lack of tonic signaling, it is difficult to appreciate how the two CD44v6-CARs may act therapeutically. And whether measuring cell size, IFN γ production in absence of antigen, or other readouts will predict therapeutically potent CARs.

Reviewer #2 (Remarks to the Author):

The authors have addressed my comments.

Reviewer #3 (Remarks to the Author):

Thank you for addressing my comments and concerns. I am satisfied with the changes made and the revised manuscript is easier to follow. I have no further concerns or comments.

Reviewer #1 (Remarks to the Author):

In the revised manuscript by Saren et al, the authors have addressed several critiques made in the initial review. The overall phenomenon of tonic signaling, exhaustion, and differences in activity are appreciated through modifications to the CDR of scFvs. However, it is still unclear whether this is a general phenomenon that requires solving for future CAR development. It is also noteworthy that several other IL13Ra2-CARs and CD44v6-CARs have been developed that lacked this type of tonic signaling, and did not necessarily require this depth in analysis. Therefore, there is still the question of whether this is a niche story in the development of one or two novel scFv-based CARs.

We would like to express our gratitude for the encouraging comment from the reviewer. We believe that antigen-independent tonic signalling is a very important aspect when developing new CARs and that understanding all potential underlying mechanisms is crucial. Previously only the framework of the scFv has been found to impact tonic signalling (PMID: 33547226, 25939063). This is not surprising as the framework sequence is important for the stability of the scFv. We report that the CDR regions can also cause antigen-independent tonic signalling of CARs. We believe that this is a general phenomenon that scientists must be aware of, but that may not always require a solution in case many scFv clones are available for further development of CAR-T cells. When we screen our IL13R α 2-CARs and CD44v6-CARs the different clones display different levels of tonic signalling, which importantly required screening at the CAR-T cell level. We also agree that in-depth analysis is probably not needed and thus we proposed a quick screening using cell size and IFN- γ , this could be used to select clones that display no or a low level of tonic signalling for further development.

1. Please explain why excess IL-13 was able to block scFv binding to IL13Ra2 with A-C, but not with D-E? This is a phenomenon that would likely partially explain antigen-dependent differences that may confound the antigen-independent exhaustion from tonic signaling. While these data were included in the initial submission, it was rewritten slightly this round in a way that caught the reviewer's attention, coupled with other revisions performed to address more differences between A-C and D-E. Overall, many differences are apparent with A-C vs D-E, but it is appreciated that the authors demonstrated that mutants in the CDR of the scFvs could reduce tonic signaling and antigen-dependent stimulation of the CAR T cells.

We thank the reviewer for allowing us to explain this better. We have performed epitope mapping of scFv B-E (not included in the paper), which revealed that scFv[B] and scFv[C] bind in the ligand binding site and thereby compete with IL-13 for binding. ScFv[D] bind closely to the site of scFv[B]-[C] but its binding is not disrupted by IL-13. scFv[E] has a unique epitope binding site away for the ligand-binding domain. We acknowledge that the difference in epitope binding might partially influence antigen-dependent differences. However, we think this does not explain the differences that we see in our study. Our data suggest that the on-target effect is strongly influenced by the tonic signalling in CAR-T cells in the unstimulated state (without antigen), and that excessive tonic signalling makes the CAR-T cells unable to respond properly when they meet target cells. This is evident in the poor proliferative ability, cytotoxic capacity, cytokine secretion and upregulation of

activation markers upon antigen stimulation. Both scFv[B] and scFv[C] have similar binding sites on IL13R α 2, despite this, CAR[B]-Ts perform poorly compared to CAR[C]-Ts. This can be explained by the antigen-independent tonic signalling observed for CAR[B]-T. Importantly, CAR[B]-Ts also go through AICD in culture, which would not be a desirable phenomenon when developing a commercial CAR-T cell product.

2. In the revised *in vivo* data (fig S3), CAR-D shows very little efficacy, compared with CAR-E. Yet both, to the extent that they were compared *in vitro*, showed little tonic signaling and comparable anti-tumor activity (except for lower IL-2 expression, fig. S2h-i, and higher Pd1 expression, S6h). Again, making the reviewer wonder whether tonic signaling is the major aspect contributing to differences between the 5 scFvs.

We agree with the reviewer that although antigen-independent tonic signalling is the major contributor to the differences observed in this study there might also be other things that can contribute, including binding affinity and target epitope.

However, we would like to emphasize that although CAR[D]-T and CAR[E]-T display a similar activity upon antigen stimulation *in vitro* there are still some differences in the unstimulated state that indicate that CAR[D]-T have slightly higher degree of tonic signalling compared to CAR[E]-T. Unstimulated CAR[D]-T has a slightly larger cell size (Fig. S6B), IFN- γ secretion (Fig. S6C), Fas expression (Fig. S6L) and also decrease in culture more rapidly than CAR[E]-T after expansion (Fig. S6N). Therefore, we believe that the differences observed between CAR[D]-T and CAR[E]-T efficacy *in vivo* is still a result of the slightly higher antigen-independent tonic signalling observed in CAR[D]-T.

3. The reviewer appreciates the inclusion of CD44v6 scFvs and clustering phenomenon between two CARs. However, affinities were not measured, and with the variations with *in vivo* therapy using IL13R α 2-CAR-D and CAR-E, which both showed a lack of tonic signaling, it is difficult to appreciate how the two CD44v6-CARs may act therapeutically. And whether measuring cell size, IFN γ production in absence of antigen, or other readouts will predict therapeutically potent CARs.

We thank the reviewer for the positive comment. Although not displayed in the manuscript the binding affinity of clone [1] and [2] in an antibody format were very similar (KD (M) = 1.1×10^{-9} and 1.4×10^{-9} respectively). Further, we agree that it would be of interest in the future to assess therapeutic efficacy of these two clones. However, we would like to point out that this manuscript is not focused on the therapeutic aspects but aims at highlighting the involvement of CDRs in CAR clustering and subsequent antigen-independent tonic signalling. We focused on the technical aspects of tonic signalling as it was pointed out in the initial submission by the editor and reviewers that our manuscript contained two stories, one more technical about tonic signalling and one therapeutic for glioblastoma therapy. Therefore, we did not investigate the *in vivo* therapeutic aspects further.

We would also like to point out that we believe that CAR[D] showed a slight degree of tonic signalling, as discussed in the response to your question #2, which can be predicted by cell

size and IFN- γ secretion from unstimulated CAR[D]-T cells. Therefore, we also believe that this can predict therapeutic potential.

Reviewer #2 (Remarks to the Author):

The authors have addressed my comments.

We thank the reviewer for the thorough revision of the manuscript and are appreciative that there are no more concerns.

Reviewer #3 (Remarks to the Author):

Thank you for addressing my comments and concerns. I am satisfied with the changes made and the revised manuscript is easier to follow. I have no further concerns or comments.

We thank the reviewer for the thorough revision of the manuscript and are appreciative that there are no more concerns.